# STING-dependent paracriny shapes apoptotic priming of breast tumors in response to anti-mitotic treatment

Steven Lohard[1,2], Nathalie Bourgeois[1,2,3], Laurent Maillet[1,2], Fabien Gautier[1,2,3], Aurélie Fétiveau[1,2], Hamza Lasla[2,3], Frédérique Nguyen [1,4], Céline Vuillier[1,2], Alison Dumont[1,2], Agnès Moreau-Aubry[1], Morgane Frapin[5], Laurent David [6,7], Delphine Loussouarn[8], Olivier Kerdraon[2,3], Mario Campone[1,2,3], Pascal Jézéquel[1,2,3], Philippe P. Juin [1,2,3]* & Sophie Barillé-Nion [1,2]*

A fascinating but uncharacterized action of antimitotic chemotherapy is to collectively prime cancer cells to apoptotic mitochondrial outer membrane permeabilization (MOMP), while impacting only on cycling cell subsets. Here, we show that a proapoptotic secretory phenotype is induced by activation of cGAS/STING in cancer cells that are hit by antimitotic treatment, accumulate micronuclei and maintain mitochondrial integrity despite intrinsic apoptotic pressure. Organotypic cultures of primary human breast tumors and patient-derived xenografts sensitive to paclitaxel exhibit gene expression signatures typical of type I IFN and TNFα exposure. These cytokines induced by cGAS/STING activation trigger NOXA expression in neighboring cells and render them acutely sensitive to BCL-xL inhibition. cGAS/STING-dependent apoptotic effects are required for paclitaxel response in vivo, and they are amplified by sequential, but not synchronous, administration of BH3 mimetics. Thus anti-mitotic agents propagate apoptotic priming across heterogeneously sensitive cancer cells through cytosolic DNA sensing pathway-dependent extracellular signals, exploitable by delayed MOMP targeting.

[1] CRCINA, INSERM, Université d'Angers, Université de Nantes, Nantes, France. [2] SIRIC ILIAD, Nantes, Angers, France. [3] Institut de Cancérologie de l'Ouest, 15 Rue André Boquel, 49055 Angers, Pays de la Loire, France. [4] Oniris, site Chantrerie, CS40706, 44307 Cedex 3Nantes, France. [5] UMR 1280 PhAN, Université de Nantes, INRA, Nantes, France. [6] Nantes Université, CHU Nantes, Inserm, CRTI, UMR 1064, ITUN, Nantes, France. [7] Nantes Université, CHU Nantes, Inserm, CNRS, SFR Santé, FED 4203, Inserm UMS 016, CNRS UMS 3556 Nantes, France. [8] Service d'Anatomie Pathologique, CHU Nantes, Nantes, France. *email: philippe.juin@univ-nantes.fr; sophie.barille@inserm.fr

Taxanes (e.g., paclitaxel) are part of the chemotherapeutic regimen widely used in the treatment of breast and other cancers. Despite some clinical efficiency, the maintenance of residual cancer cell subsets after antimitotic treatment remains a major conundrum. Fractional killing may result from the existence of distinct subclones and/or from the diversity of individual responses from populations of the same progeny. There is indeed a tremendous heterogeneity in cancer cell fates following exposure to paclitaxel and blocking of mitotic progression[1]. Single cell variability in paclitaxel response ensues from: (i) individual cell cycle progression during the time of drug exposure; (ii) a competitive molecular race between cell death and mitotic slippage signaling networks upon mitotic arrest; (iii) multinucleation and resumption of transcriptional programs altering cell survival beyond slippage. In all conditions, it is understood that stochastic variations in key actors of instrumental pathways contribute to fractional response and, by inference, that reducing intrinsic resistance pathways would improve the global response[2,3].

Radiotherapy, chemotherapy, and immunotherapy promote cancer cell apoptosis, a mode of cell death executed by caspases activated downstream of mitochondrial outer membrane permeabilization (MOMP)[4,5]. MOMP is regulated by a network of intracellular interactions through which anti-apoptotic BCL-2 homologs (BCL-xL, MCL-1) bind and inhibit functionally distinct, complementary proapoptotic counterparts (multidomain BAX/BAK and their upstream regulatory BH3-only proteins such as BID, BIM, or NOXA). Antimitotic treatment leads to many phosphorylation/dephosphorylation and degradation reactions that impact the BCL-2 network and favor MOMP during arrest and slippage[6–8]. Current data indicate that cancer cells treated by antimitotic drugs accumulate intrinsic proapoptotic signals upstream of MOMP that render BCL-xL particularly required for survival maintenance[7,9,10]. This has two implications. Firstly, the predisposition of a significant fraction of cancer cells to undergo MOMP-induced apoptosis (in a BCL-xL sensitive manner), should be necessary for human tumors to respond to paclitaxel, as reported for other chemotherapeutic or targeted therapies[5,11]. Secondly, enhancing "apoptotic priming" (that is, cancer cell propensity to undergo MOMP) should improve population response to antimitotic treatment. Direct targeting of MOMP can now be achieved by selective BH3 mimetic inhibitors of interactions between anti and proapoptotic BCL-2 family members. Venetoclax (selectively targeting BCL-2) was the first BH3 mimetic approved for treating BCL-2-dependent chronic lymphocytic leukemia[12]. Dual BCL-2/BCL-xL inhibitors (ABT-737 or ABT-263/Navitoclax) or selective inhibitor of BCL-xL (WEHI-539) are also available and some have already been shown to potentiate the effects of chemotherapy in breast cancers in preclinical studies[13–15].

Current attempts to improve antimitotic treatment efficiency with BH3 mimetics intend to circumvent the effects of cell to cell variations by enhancing cell death rates in cell populations. Whether and how BH3 mimetics might collectively improve the responses to antimitotic therapy of breast cancer cell populations with heterogeneous proliferative rates remains nevertheless uncharacterized. To address this issue, there is a need to unravel mechanisms by which antimitotic drugs may promote proapoptotic signals in an entire breast cancer population while only impacting the cycling cell subset. We herein explore what role the cGAS/STING pathway may play in the induction of apoptotic priming by antimitotic treatment. Indeed, such treatment promotes the occurrence of micronuclei in proliferating cells[16], micronuclei are prone to activate this cytosolic DNA sensing pathway and the latter influences intercellular communications[17]. Using human samples manipulated ex vivo, in vivo models of patient-derived xenografts and genetically engineered cell lines, we reveal that paclitaxel treatment enhances BCL-xL apoptotic priming not only by intrinsic signals but also by extrinsic ones relying on type I interferon and TNFα produced by cGAS/STING activation in proliferative subpopulations. Consequently, BH3 mimetics targeting BCL-xL improve in vivo response to antimitotics if their administration allows STING-active cancer cells to mount a transcription-dependent paracrine effect before killing them.

## Results

**Paclitaxel treatment triggers a proapoptotic secretome**. Two downstream markers of STING activation, namely activating phosphorylations of STAT1 (pTyr701-STAT1) and of NF-κB (pSer536-p65) were induced in breast cancer cell lines treated with the antimitotic agent paclitaxel in vitro (Fig. 1a and Supplementary Fig. 1a). These responses were lost following CRIPSR-mediated STING knock-out (KO) in breast cancer cell lines. Arguing for a critical upstream role for cytosolic DNA, knocking out the sensor cGAS also dramatically decreased STAT1 response (Fig. 1b and Supplementary Fig. 1b). We assessed whether paclitaxel-induced activation of STING ensued from the formation of micronuclei activating cGAS. In vitro experiments revealed cGAS-bound micronuclei 24 h after paclitaxel treatment (Fig. 1c). Moreover, cGAS-positive micronuclei were also observed after in vivo paclitaxel treatment of human breast cancer patient-derived xenografts (PDX) grown in immunodeficient mice (described in Supplementary Fig. 3a) (Fig. 1d). Indicating that cGAS recruitment to micronuclei and subsequent STING activation is due to cytosolic exposure of micronuclei DNA, overexpression of LaminB2 (LMNB2) in cancer cell lines, shown to prevent this process[17], inhibited cGAS micronuclear localization following treatment (without influencing micronuclei formation per se) and prevented STAT1 and NF-κB activation (Fig. 1c, e). In contrast, mitochondrial DNA (mtDNA) release following MOMP, that might contribute to cGAS/STING pathway activation[18,19], did not play a significant role since BAX/BAK double KO cells did not prevent STAT1 or NF-κB activation by paclitaxel treatment (Supplementary Fig. 1c).

The cGAS/STING cytosolic DNA sensing pathway pleiotropically affects intercellular communications. We thus investigated whether its specific activation by paclitaxel modified the secretory phenotype of breast cancer cells, possibly leading to the propagation of apoptotic signals between cells. To this end we prepared media conditioned by "donor" cancer cells (id est transiently exposed to paclitaxel for 24 h or not, washed out and left untreated for an extra 2 days prior media collection). To evaluate the proapoptotic effects of these conditioned media (CM) and/or their ability to enhance apoptotic pressure on specific antiapoptotic proteins, we added them to "recipient" cancer cells alone or in combination with distinct BH3 mimetics targeting either BCL-2 (ABT-199), BCL-xL (WEHI-539), or MCL-1 (S63845) prior evaluation of cell death rates. CM from paclitaxel-treated donors strongly increased BCL-xL apoptotic priming in recipients, as they potently and specifically sensitized them to treatment by WEHI-539 (but neither to ABT-199 nor to S63845 (Fig. 1f)) in a pan-caspase inhibitor sensitive manner (Supplementary Fig. 1d). Clonogenic assays confirmed long lasting effects of CM combined with BCL-xL inhibition (Supplementary Fig. 1e). Induction of BCL-xL dependency by the paracrine effects of paclitaxel treatment was also detected in the non small cell lung cancer (A549) or ovarian cancer (SK-OV-3) cell lines (Supplementary Fig. 1f, g). Importantly, either STING or cGAS KO or LMNB2 overexpression in donor breast cancer cells strongly decreased induction of paracrine propapoptotic

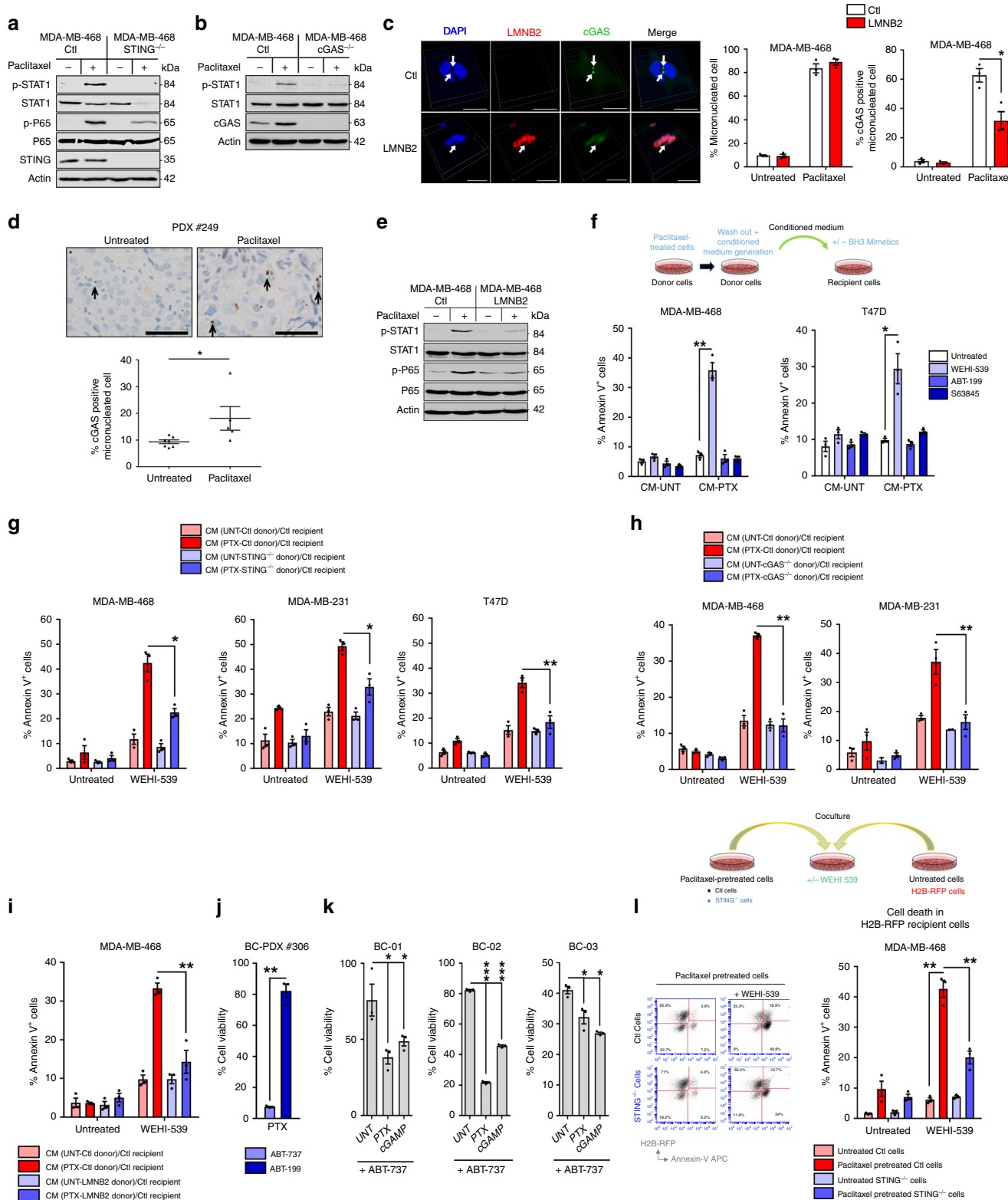

effect by paclitaxel (Fig. 1g–i and Supplementary Fig. 1h). We note that in comparison, deleting STING in recipient cells had no impact (Supplementary Fig. 1i). In contrast, CM from BAX/BAK double KO donor cells were as efficient as those of control donors to promote apoptosis, arguing again that mtDNA did not play a significant role in this effect (Supplementary Fig. 1j).

To corroborate that STING activation contributes to enhancement of apoptotic priming by paclitaxel treatment also in

primary breast cancer cells, we used organoids derived either from PDX or from freshly excised human breast cancer specimen (Patient-Derived Organoids PDO) where synergistic effects on cell viability between paclitaxel and ABT-737 (but not ABT-199) were detected (Fig. 1j). The STING agonist cGAMP also sensitized PDO to ABT-737 (Fig. 1k).

In another series of experiments, cancer cell lines that were previously treated by paclitaxel were directly put in contact to

**Fig. 1 a STING-dependent paclitaxel-induced secretome spreads BCL-xL apoptotic priming. a, b** STAT1 and NF-κB pathway analysis of 24 h-paclitaxel treated or not STING$^{-/-}$ (**a**) and cGAS$^{-/-}$ (**b**) or control MDA-MB-468 cells by immunoblotting. **c** Representative pictures of cGAS immunostaining (left panel, scale bar: 20 μm) and quantification of micronucleated cells (middle panel) and cGAS-positive micronuclei (left panel) in control or LMNB2 overexpressing MDA-MB-468 cells after paclitaxel treatment. **d** cGAS-positive micronucleated cells quantification in paclitaxel-treated or not in PDX #248 and PDX#249 (lower panel) and representative fields after IHC staining with anti-cGAS (upper panel; scale bar: 50 μm). **e** STAT1 and NF-κB pathway immunoblot analysis in paclitaxel-treated or not, LMNB2 overexpressing or control MDA-MB-468 cells. **f** 24 h-paclitaxel treated or not (donor) cells were washed out to produce 48 h-conditioned media (CM) that were applied to untreated (recipient) corresponding cancer cells for 48 h in presence or not of the BH3 mimetics WEHI-539, ABT-199, or S63845. Apoptotic index in recipient breast cancer cells was assessed using Annexin-V staining. **g–i** Same experiment as (**f**) using control or STING$^{-/-}$ (**g**) or cGAS$^{-/-}$ (**h**) or LMNB2 overexpressing (**i**) indicated donor cells in presence of the BH3 mimetic WEHI-539 or not. **j** Cell viability of PDX #306 cultured in organoids (BC-PDX#306) in paclitaxel-treated conditions for 4 days and ABT-737 or ABT-199 BH3 mimetics added for the 2 last days (sequential treatment). **k** Cell viability of 3 distinct patient-derived organoids (BC-01, BC-02, and BC-03) in ABT-737-treated conditions plus paclitaxel or cGAMP. **l** 24 h-paclitaxel-pretreated control or STING$^{-/-}$ MDA-MB-468 cells were cultured with untreated MDA-MB-468 H2B-RFP expressing cells. After 48 h, co-cultures were treated or not with WEHI-539 for additional 48 h and cell death was assessed in each cell population. Error bars indicate mean $+/-$ SEM; Two-sided paired $t$-test. The symbols correspond to a $p$-value inferior to *0.05, **0.01, and ***0.001. NS: not significant. Data were collected from $n = 3$ independent experiments except **d** ($n = 5$).

untreated cell lines expressing H2B-RFP (used as a discrimination marker). These assays confirmed less efficient sensitization to WEHI-539 of RFP-positive cells in contact with STING-depleted compared to these in contact to wild-type cells (Fig. 1l and Supplementary Fig. 1k).

Cycling of donor cells was required for paclitaxel treatment to induce pro-apoptotic paracrine signals, since CM from serum-deprived (low cycling) or thymidine-blocked paclitaxel-treated donors were inefficient (Supplementary Fig. 1l, m). Of note paclitaxel-treated CM did not alter recipients' cell cycle, ruling out the presence of residual paclitaxel in CM (Supplementary Fig. 1n). Another antimitotic agent, the Aurora-B inhibitor AZD1152 also induced micronuclei and paracrine proapoptotic effects, in contrast to etoposide, even though this genotoxic agent was directly cytotoxic (Supplementary Fig. 1o, p).

Altogether, these results indicate that paclitaxel treatment recruits cGAS/STING activation in response to unstable nuclear membrane of induced micronuclei and that this induces a secretory phenotype which promotes BCL-xL-dependent apoptotic priming in untreated cancer cells.

**IFN-I/TNF signatures in paclitaxel sensitive breast tumors.** Functional assays of numerous patient derived samples allowed us to hint on the molecular basis of the pro-apoptotic paracrine effects of paclitaxel treatment reported above. As previously described[20], we explored the apoptotic response to paclitaxel and to ABT-737 of 163 breast tumor samples freshly obtained from patients who underwent surgical excision and processed in 3D organotypic ex vivo culture for 2 days after tumor slicing (cohort described in Supplementary Fig. 2a). Comparison of apoptotic rates in cancer cells by immunohistochemistry (IHC) analysis of tumor slices exposed for 48 h to compounds and in adjacent untreated control slices (using active caspase-3 as a marker), showed great inter-patient heterogeneity of responses (Fig. 2a). The great majority of tumors showing paclitaxel sensitivity (that is, more than 20% cell death above control) were sensitive to induction of cancer cell death by ABT-737 (Fig. 2b). This indicates that cancer cell apoptotic priming (on BCL-2, BCL-xL, or both) is necessary to acute paclitaxel sensitivity and it is consistent with the notion that paclitaxel enhances BCL-xL dependency. Conversely, however, a large part of ABT-737 sensitive tumors did not detectably respond to paclitaxel in our organotypic assay. This indicates that BCL-2/BCL-xL-dependent apoptotic priming is not sufficient to warrant acute sensitivity to paclitaxel, and that additional features are required. Differing proliferation rates did not account for the differences in sensitivity to paclitaxel in apoptotic primed tumors as Ki-67 staining neither correlated nor predicted paclitaxel sensitivity in these tumors (Fig. 2c) and was not significantly different between sensitive or resistant tumors (Fig. 2d).

To identify what determines paclitaxel sensitivity in addition to apoptotic priming, we analyzed the transcriptomes of 45 paclitaxel resistant and 15 sensitive tumors (detailed in Supplementary Fig. 2b, c) and evaluated gene expression signatures of a range of signaling pathways and cancer phenotypes. Paclitaxel sensitive tumors displayed significantly higher scores for basal STAT1 and type I IFN signaling pathways but no differences in proliferation scores (Fig. 2e, f). We measured concentrations of IFNα in available media from adjacent tumor slices treated or not by paclitaxel (7 responders and 6 non responders). Basal IFNα production was not significantly different between untreated paclitaxel sensitive or resistant tumor supernatants. However, increased IFNα secretion after paclitaxel treatment was preferentially detected in sensitive tumors compared to the resistant ones (Fig. 2g). Thus, not only cancer cell apoptotic primed state but also an actionable type I IFN signaling pathway are associated with paclitaxel cytotoxic efficacy in breast tumors.

The absence of significant differences in immune and stromal scores between paclitaxel sensitive and resistant samples (Supplementary Fig. 2d), suggests that immune cells do not necessarily (or not only) contribute to type I IFN signaling during paclitaxel response. This was further supported by our investigation of the in vivo response to paclitaxel of 3 different human breast cancers grown as patient-derived xenografts (PDX) in immunodeficient mice (described in Supplementary Fig. 3a). *IFNB1* mRNA evaluation by qPCR revealed that type I IFN pathway was readily activated upon paclitaxel treatment in the 3 PDX models (Fig. 3a). This coincided with an antitumor activity of paclitaxel in these PDX (Supplementary Fig. 3b). Consistently, transcriptomic analysis of these tumors achieved using DGE-RNAseq and Omic tool Enrichr revealed gene expression changes typically associated with the response of mammary epithelial cells to inflammatory cytokines (Fig. 3b). Importantly, this analysis hinted on the concurrent activation of TNF signaling, and *TNF* mRNA expression was induced in the 3 PDX after treatment by paclitaxel (Fig. 3c), as well as type I IFN target genes (Supplementary Fig. 3c). This strongly argues for the triggering of a complex inflammatory signaling upon paclitaxel treatment involving TNF and type I interferons.

**Paracrine IFN-I and TNFα promote NOXA-dependent apoptosis.** The above results led us to investigate whether the paracrine proapoptotic effects of paclitaxel involved type I IFN and/or TNFα. Corroborating the findings on ex vivo and PDX results we observed transcriptional induction of *TNF* and *IFNB1*, in cells

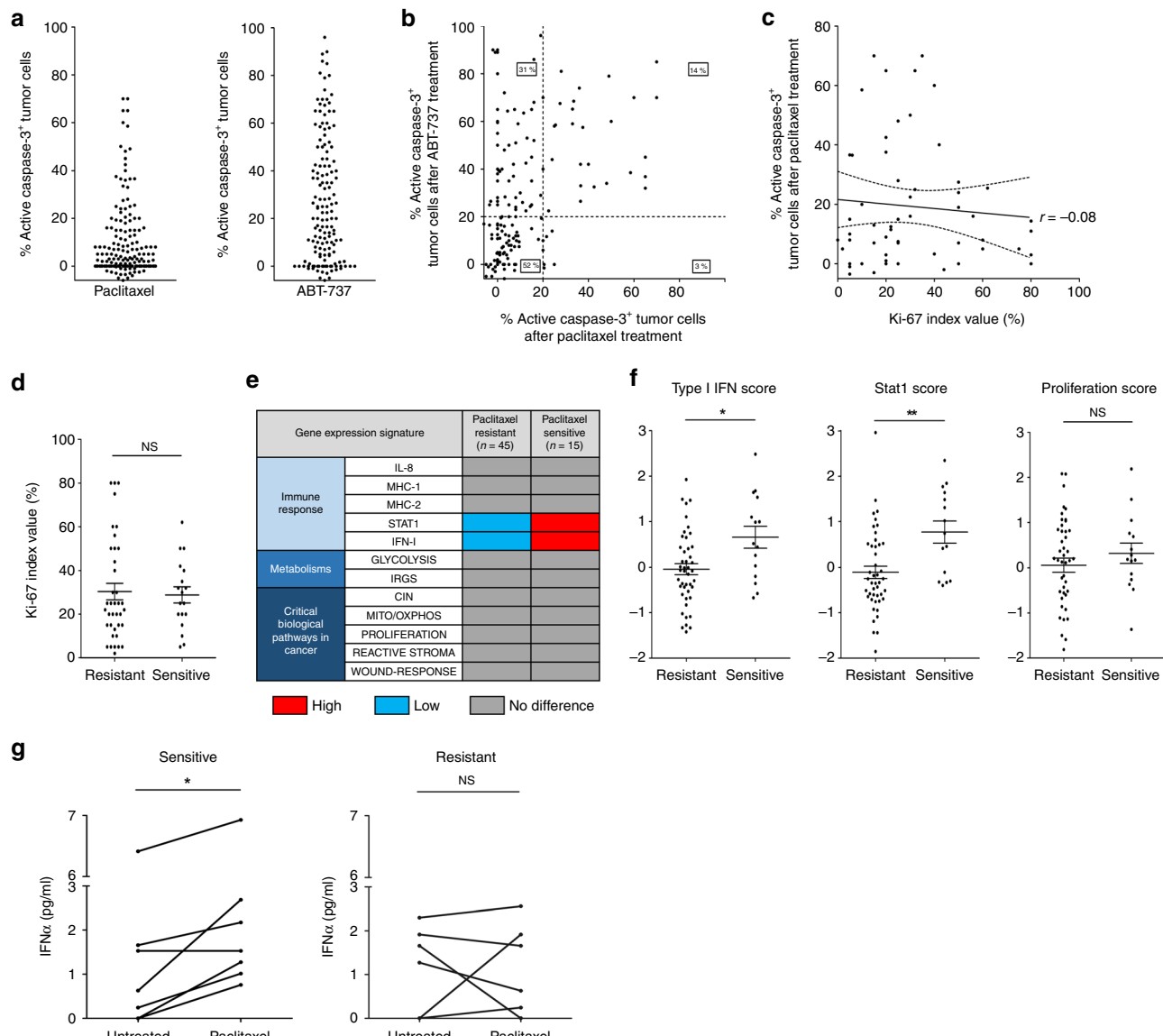

**Fig. 2 Type I IFN signature and high apoptotic primed state coincide with breast tumor ex vivo sensitivity to paclitaxel. a, b** Apoptotic index in 163 fresh human breast tumors defined by the % of active caspase-3 positive tumor cells detected by IHC after 48 h ex vivo paclitaxel (left) or ABT-737 (right) treatments separately (**a**) and co-response (**b**). **c, d** Tumor Ki-67 index (%) in 56 ABT-737 sensitive tumors (above 20% of active caspase-3 positive tumor cells) related to the % of active caspase-3 positive tumor after paclitaxel treatment **c** or related to paclitaxel sensitivity/resistance (20% apoptotic index threshold) (**d**). **e** Comparison of gene expression signatures between both paclitaxel resistant and paclitaxel sensitive tumor groups (60 classified tumors based on their ex vivo paclitaxel sensitivity). **f** STAT1, type I IFN and proliferation scores obtained from the transcriptome. **g** IFNα quantification in supernatants of tumor slices treated or not with paclitaxel. $n = 5$ sensitive tumors and $n = 6$ resistant tumors. Error bars indicate mean $+/-$ SEM; Two-sided unpaired (**d, f**) and paired (**g**) $t$-test. The symbols correspond to a $p$-value inferior to *0.05 and **0.01. NS: not significant.

exposed to paclitaxel in vitro that was inhibited by *LMNB2* overexpression (Fig. 4a) and STING KO (Fig. 4b and Supplementary Fig. 4a) or cGAS KO (Fig. 4c). In addition, STING KO also abrogated IFNα production after paclitaxel treatment (Fig. 4d and Supplementary Fig. 4b). These results underscore a role for unstable micronuclei-induced STING activation. We could not formally incriminate STING in the inflammatory effects of in vivo paclitaxel treatment of PDX, but we confirmed that ex vivo treatment of 2 PDX-derived organoids with the STING agonist diABZI[21] phenocopied paclitaxel ability to induce *TNF* and *IFNB1* (Fig. 4e).

To assess the functional importance of type I IFN and TNFα signaling in paclitaxel-induced paracrine effects, we used IFNα/β receptor alpha chain (*IFNAR1*) KO cell lines as recipients or either *TNF* KO or NF-κB super repressor mutated-IκBαSR (IκBαSR) cells as donors. Either manipulation alone inhibited paracrine apoptotic priming by paclitaxel (Fig. 4f, g and Supplementary Fig. 4c–e). Combination of *TNF* KO donor cells and *IFNAR1* KO recipient ones did not provide further protection, arguing that type I IFN and TNFα act in concert to promote apoptotic priming (Supplementary Fig. 4f). Accordingly, addition of inhibitory anti-TNFα to CM from paclitaxel-treated wild-type donors strongly decreased their proapoptotic effects (Fig. 4h). Recombinant TNFα enhanced sensitivity of breast cancer cells to WEHI-539 and this was further enhanced in the presence of recombinant IFNα (Fig. 4i and Supplementary

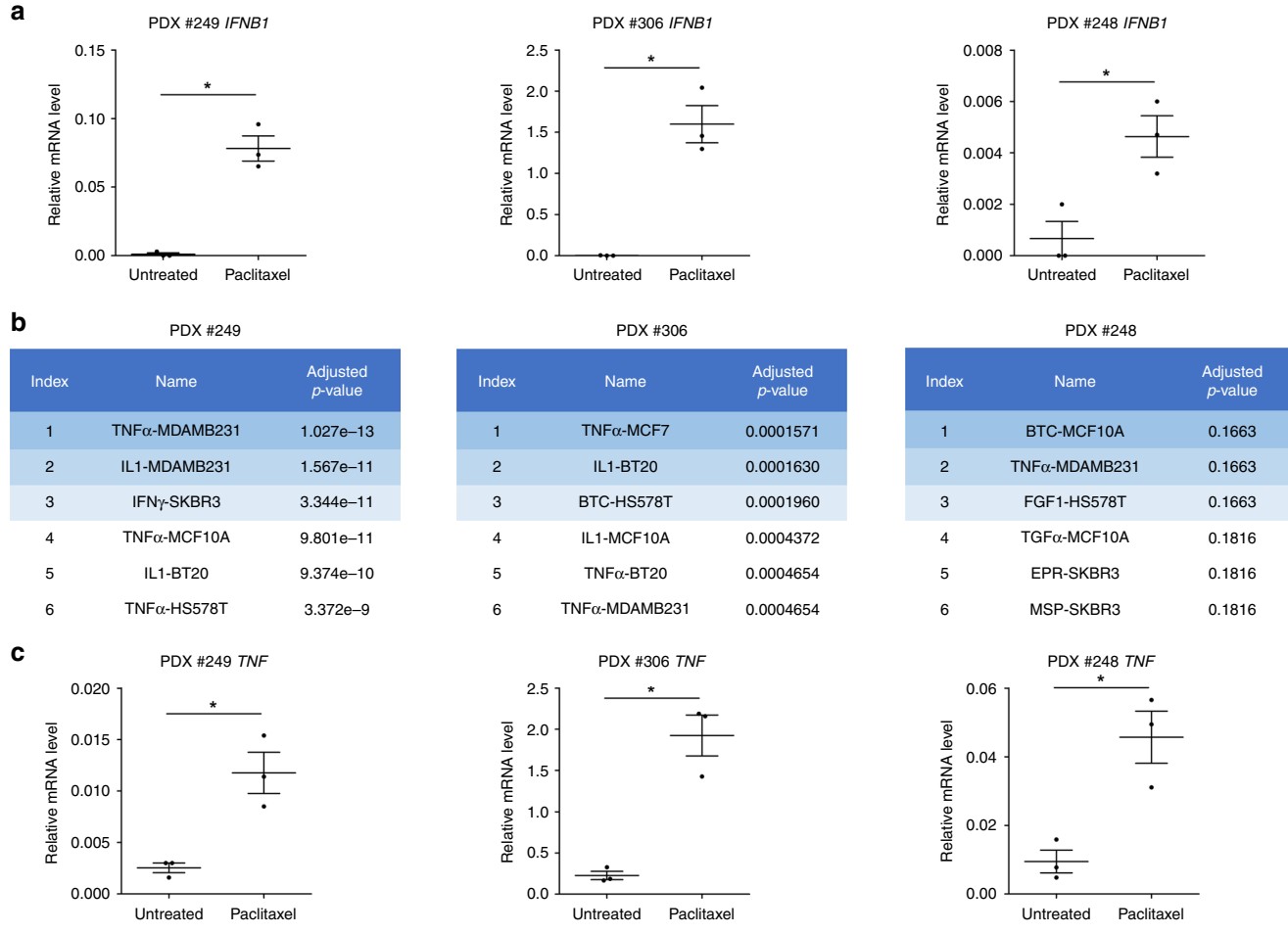

**Fig. 3 Paclitaxel-treated cancer cells activate type I IFN and TNFα through the cGAS/STING pathway. a** *IFNB1* qPCR analysis in PDX treated or not by paclitaxel. **b** Global transcriptomic analysis in paclitaxel-treated PDX compared to untreated PDX using Enrichr omic tool LINCS L1000 Ligand Perturbations up. **c** *TNF* qPCR analysis in paclitaxel-treated or untreated PDX. $n = 3$ mice per group, Error bars indicate mean $+/-$ SEM; Two-sided paired *t*-test. The symbols correspond to a *p*-value inferior to *0.05.

Fig. 4g). Altogether, these results argue for a synergistic bystander effect of soluble TNFα and type I IFN produced by STING-active donor cells in response to paclitaxel treatment.

To define the molecular mechanisms involved in paclitaxel induction of proapoptotic paracrine effects, we evaluated BCL-2 family member expression changes in breast cancer cell lines exposed to CM from paclitaxel-treated donors. Immunoblot analysis indicated that cancer cells receiving CM from paclitaxel-treated donors displayed an increased and long lasting expression of NOXA, a BH3-only protein essentially functioning as an endogenous MCL-1 inhibitor. In contrast the expression of other BH3-only pro-apoptotic BCL-2 family proteins, BID, BIM, or PUMA remained unchanged (Fig. 5a). NOXA/*PMAIP1* mRNA was also increased in recipient cells following exposure to CM from paclitaxel-treated donors but not BID, BIM (*BCL2L11*) or PUMA (*BBC3*) mRNAs (Fig. 5b and Supplementary Fig. 5a). This coincided with decreased MCL-1 protein (but not mRNA) expression and with unchanged BCL-2 or BCL-xL protein and mRNA expressions. NOXA protein and mRNA expressions were also induced in PDX models following paclitaxel treatment (Fig. 5c). Importantly, knocking out NOXA/*PMAIP1* in recipient cells completely blocked apoptotic priming by CM from paclitaxel-treated donors (Fig. 5d), in contrast to BIM or BID KO (Supplementary Fig. 5b). In the same experimental setting, we showed that BAX/BAK double KO in recipient cells completely

prevented their apoptosis commitment (Supplementary Fig. 5c). Importantly, NOXA-induced expression in recipient cells relied on STING and TNFα expression in donors, as well as IFNAR1 expression in recipient cells (Fig. 5e and Supplementary Fig. 5d). Accordingly, recombinant TNFα/IFNα combination strongly synergized to specifically promote NOXA protein and *PMAIP1* mRNA expression (Fig. 5f and Supplementary Fig. 5e) and knocking out *PMAIP1* gene completely prevented apoptotic priming by IFNα/TNFα exposure (Fig. 5g and Supplementary Fig. 5f). Consistent with the notion that NOXA expression contributes to enhance apoptotic priming on BCL-xL by antagonizing the complementary activity of MCL-1, BCL-xL KO in recipient cells enhanced apoptotic priming effect by CM from paclitaxel-treated donors, and by the TNFα/IFNα combination (Supplementary Fig. 5g) while MCL-1 KO did not, even though it dramatically sensitized cells to WEHI-539 (Supplementary Fig. 5h).

Transcriptional induction of NOXA by type I IFN may result from the activation of IFN-stimulated response elements in the regulatory region of the human *PMAIP1* gene[22]. ChIP assay indeed revealed recruitment of IRF3 to the PMAIP1 promoter in paclitaxel or TNFα/IFNα-treated cells (Supplementary Fig. 5i) and consistently, knocking down IRF3 using RNA interference prevented NOXA expression and apoptotic priming in cells receiving CM from paclitaxel-treated donors or TNFα/IFNα treatment (Supplementary Fig. 5j, k).

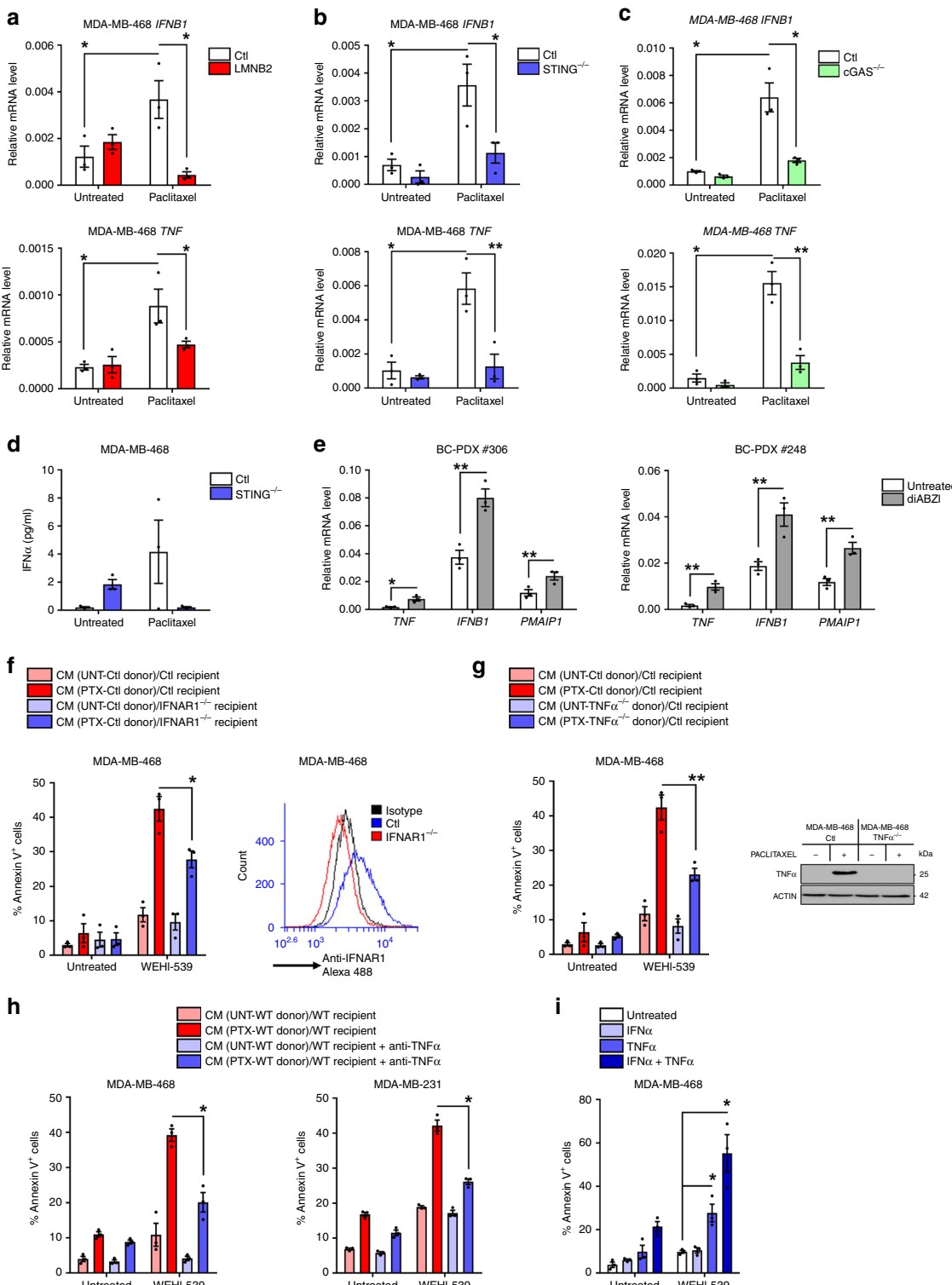

**Fig. 4 STING-dependent induction of TNFα and type I IFN triggers a proapoptotic secretome. a–c** *IFNB1* and *TNF* qPCR analysis in LMNB2 overexpressing (**a**), STING $^{-/-}$ (**b**) or cGAS $^{-/-}$ (**c**) and control MDA-MB-468 cells after 24 h paclitaxel treatment. **d** IFNα production before and after paclitaxel treatment in control or STING$^{-/-}$ MDA-MB-468 cells. **e** *TNF, IFNB1* and *PMAIP1* qPCR analysis in BC-PDX #306 and BC-PDX #248 after a 48 h-treatment by the STING agonist diABZI. **f** 24 h-paclitaxel-treated or not (donor) cells were washed out to produce 48 h-conditioned media (CM) that were applied to untreated (recipient) IFNAR1 $^{-/-}$ cancer cells for 48 h in presence or not of the BH3 mimetic WEHI-539. Apoptotic index in recipient breast cancer cells was assessed using Annexin-V staining. IFNAR1 expression was detected by cytometry after immunostaining (right panel). **g** Same experiment as (**f**) using control or TNFα$^{-/-}$ MDA-MB-468 donor cells and TNFα expression was detected by immunoblot analysis in monensin-treated cells for 72 h after 24 h paclitaxel treatment or not (right panel). **h** Same experiment as (**f**) using TNFα blocking antibody preincubated in CM before addition to MDA-MB-468 (left) or MDA-MB-231 (right) recipient cells. **i** Apoptotic effect of recombinant IFNα and/or TNFα on MDA-MB-468 cells treated or not with WEHI-539 during 48 h. Data were collected from at least $n = 3$ independent experiments. Error bars indicate mean $+/-$ SEM; Two-sided paired $t$-test. The symbols correspond to a $p$-value inferior to *0.05 and **0.01.

**Paracrine apoptotic priming determines paclitaxel response.**
Apoptotic priming intrinsically induced by paclitaxel treatment in donor cells shared many mechanistic features with that extrinsically induced in recipient cells. Paclitaxel-treated donor cells indeed showed increased NOXA expression and were dramatically sensitized to induction of cell death by WEHI-539 treatment or BCL-xL KO in a BAX/BAK and NOXA dependent manner (Fig. 6a–d). Incriminating fragile micronuclei, increased NOXA and apoptotic priming were prevented by LMNB2 overexpression in donor cells (Fig. 6e, f). One crucial difference was that STING KO cells were as sensitive to intrinsic induction of apoptotic priming and nearly as prone to NOXA induction as parental cells (Fig. 6g, h and Supplementary Fig 6a, b). Thus STING is mostly required for elaboration of non cell autonomous signals upon antimitotic therapy and dispensable for cell autonomous ones. We exploited this differential requirement to determine the contribution of paracrine signals to in vivo tumor response, using wild type and STING KO MDA-MB-231 orthotopic xenograft experiments. Strikingly, in vivo paclitaxel response was completely abolished in STING KO model compared to the control one (Fig. 7a) and STAT1 and NF-κB pathway activation or *TNF*, *IFNB1*, or NOXA/*PMAIP1* gene transcription induction were also impaired in these tumors, further arguing for a crucial role of non cell autonomous apoptotic signals in paclitaxel antitumoral efficacy (Fig. 7b). In addition, the same experiments using *cGAS* or *TNF* KO cells in vivo showed that either depletion deeply impaired paclitaxel tumor response (Fig. 7c, d).

Since paracrine effects of paclitaxel are required in vivo, and since cells that supply these effects are BCL-xL-primed themselves, improvement of the global response by BCL-xL inhibition depends on how it influences the production of death signals by donor cells. Importantly, we found that CM from paclitaxel-treated BCL-xL KO donor cells exhibited significantly decreased apoptotic priming activity. We ascribe this loss of effect to massive caspase activation and cell death rates in paclitaxel-treated BCL-xL KO donors (Supplementary Fig. 7a): when these cells were treated with the pancaspase inhibitor (Q-VD-OPh), a condition necessary to obtain sufficient material for RNA and protein analysis, we could indeed check that BCL-xL KO did not directly impair STAT1/NF-κB pathway activation and *IFNB1*/*TNF* gene expression induction by paclitaxel (Supplementary Fig. 7b, c).

We inferred from the above results that there should be an optimized timing of administration for pharmacological inhibitors of BCL-xL to fully exploit apoptotic priming induced by paclitaxel treatment in vivo. Since BCL-xL-mediated maintenance of donor cell survival is critical, the elaboration of pro-apoptotic paracrine signals by paclitaxel treatment should indeed be allowed before acute BCL-xL inhibition. Using orthotopic xenografts of wild type and STING KO MDA-MB-231 cells, we thus compared 2 therapeutic administration schemes. The first consisted in a cotreatment with paclitaxel and the clinically relevant BH3 mimetic ABT-263/Navitoclax (synchronous protocol) and the second in a sequential treatment beginning with paclitaxel before Navitoclax (sequential protocol), (Fig. 7e). Strikingly, whereas Navitoclax hardly exerted any effect upon cotreatment, it significantly enhanced paclitaxel antitumor efficacy when administered in a delayed manner (Fig. 7e). STING expression was strictly necessary for this optimized therapeutic regimen to be efficient. A sequential combination was also better than a synchronous one to increase the in vitro response in organoids obtained from one of our PDX to one dose of paclitaxel/Navitoclax (Supplementary Fig. 7d).

These data led us to investigate the apoptotic activity of a recently described synthetic small molecule diABZI STING agonist-1[21]. Importantly this compound phenocopied paclitaxel's ability to sensitize breast cancer cells lines to the BCL-xL inhibitor WEHI-539 but it did not sensitize them to the BCL-2 inhibitor ABT-199 (Fig. 7f and Supplementary Fig. 7e). Of note, and in contrast, no additive effect was detected when the STING agonist was combined to paclitaxel. Moreover, this effect was abrogated in STING KO cells demonstrating its target specificity (Fig. 7g) and severely impaired in TNF KO cells (Fig. 7g) or NF-κB repressor expressing cells (Fig. 7h and Supplementary Fig. 7f), arguing for a major role of NF-κB/TNF pathway in promoting paracrine cell death upon diABZI treatment. Accordingly, this compound increased NOXA expression in cancer cells (Fig. 7i) and lost its apoptotic priming effect effect in NOXA KO cells (Fig. 7g) as described above for paclitaxel treatment. Importantly, the STING agonist also altered cell viability in breast cancer organoid model when combined with the BH3 mimetic ABT-737 (Fig. 7j) and decreased tumor progression particularly in combination with Navitoclax in the in vivo MDA-MB-231 xenograft model (Fig. 7k).

## Discussion

Antimitotic drugs trigger intrinsic apoptotic stimuli in mitotic-arrested and in mitotic-slipped cells. We herein show that the latter cells are endowed with a proapoptotic secretome due to cGAS/STING pathway activation which spreads apoptotic pressure to cancer cells not directly affected by the drugs (e.g., non cycling during treatment). Mitotic disturbance induced by paclitaxel generates micronuclei prone to envelope collapse due to defects in nuclear lamina assembly[23]. Our detection of LMNB2 sensitive cGAS recruitment on paclitaxel-induced micronuclei and of downstream activation of the cGAS/STING pathway activation is in agreement with recent reports of cGAS activation by irradiation-induced micronuclei formation[16,17]. Induction of a proapoptotic secretome is critical for in vivo response to paclitaxel as STING deletion (that does not affect the intrinsic apoptotic response to paclitaxel), that of cGAS or that of TNFα dramatically diminished it. The role of STING in the biological behavior of tumors is controversial[24,25] but it contributes to tumor growth inhibition induced by topotecan, oncolytic virus or PARP inhibitors in immunocompetent mice[26–28]. Our data establish that it plays a role in the response to chemotherapy even in the absence of immune environment by a process that relies on its modulation of cancer cell secretion. Altogether, these data predict that defects in the cGAS/STING cytotoxic pathway (or TNF/IFN pathways as discussed below) should shape tumor evolution during anti-mitotic therapy.

Soluble factors produced by paclitaxel-treated cancer cells were identified thanks to functional and molecular analysis of patient-derived organotypic ex vivo cultures and in vivo treated PDX. Sensitivity to antimitotic agents was associated with: (i) a high pre-treatment mitochondrial apoptotic priming, in agreement with[11]; (ii) additional, inducible, type I interferon and TNF responses, as previously reported for PDX treated by anthracyclin/cyclophosphamide[29] and for paclitaxel-treated MCF7 cells[30], respectively. TNFα, type I IFN and their related pathways are linked to anti-cancer therapeutic responses[31,32] and we herein establish that their direct effects on cancer cells are instrumental in addition to their modulation of cancer immune-surveillance[33,34]. TNFα/type I IFN enhancement of apoptotic priming in breast cancer recipient cells highlights a synergistic pro-death activity between these two inflammatory cytokines, recently reported to induce necroptosis in RIPK3 competent (non cancer) cells[35]. We ascribed a major role for NOXA in apoptotic priming induced by these combined cytokines. NOXA induction may ensue from a complex interaction between NF-κB and type I Interferon signaling as we

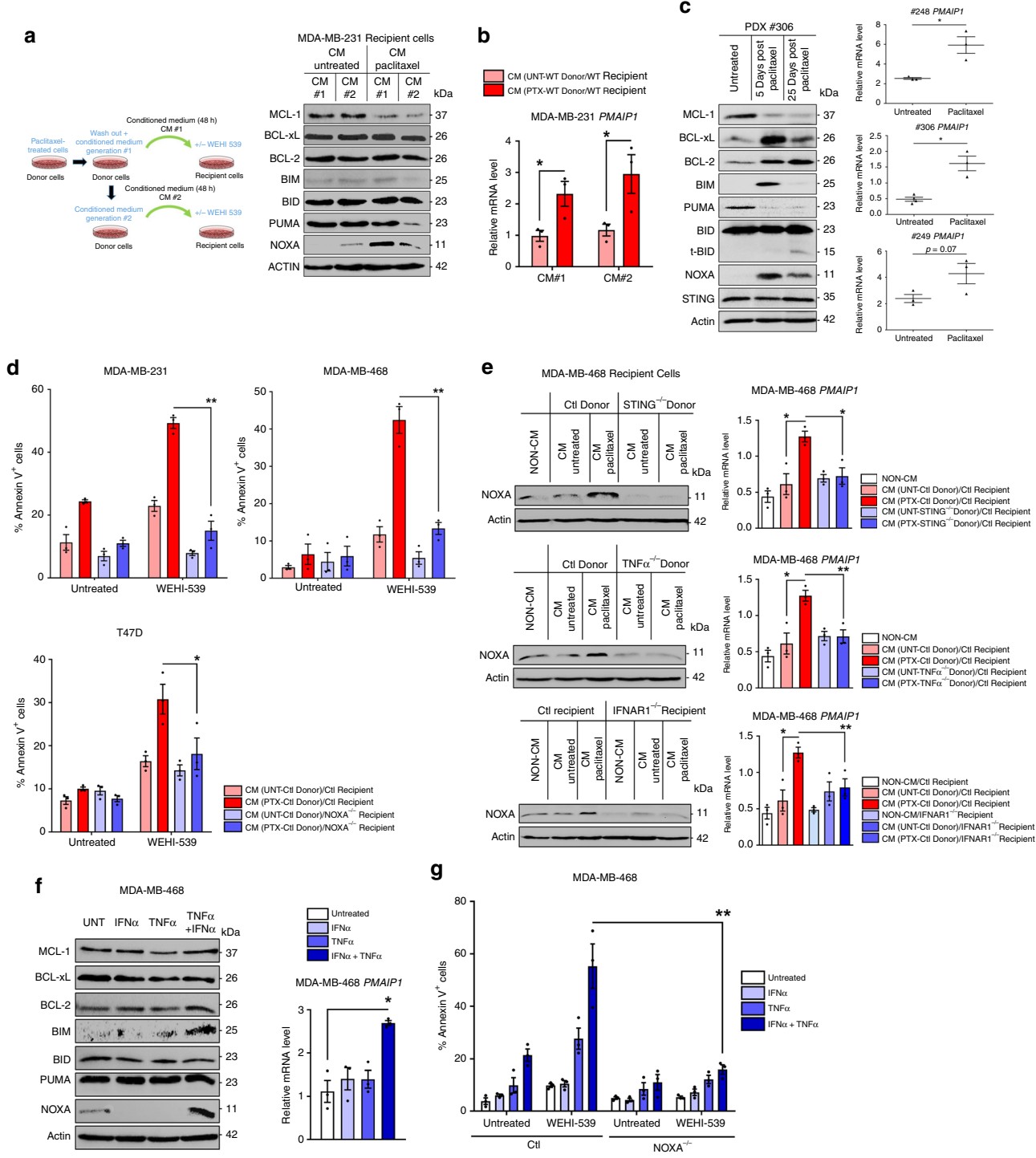

**Fig. 5 NOXA is induced by STING-dependent extrinsic paclitaxel-induced apoptotic priming signals. a, b** Immunoblot (**a**) and *PMAIP1* qPCR (**b**) analysis in MDA-MB-231 recipient cells 48 h incubated with #1 or #2 consecutively produced CM as described in the protocol depicted in the figure. **c** Immunoblot and *PMAIP1* qPCR analysis in PDX treated or not by paclitaxel after indicated times. **d** After a 24 h paclitaxel-treatment or not, (donor) breast cancer cells were washed out to produce 48 h-CM that were applied to control or NOXA$^{-/-}$ untreated (recipient) corresponding cancer cells for 48 h in presence or not of the BH3 mimetic WEHI-539. Apoptotic index in recipient breast cancer cells was assessed using Annexin-V staining. **e** NOXA immunoblot (left panel) and *PMAIP1* qPCR (right panel) analysis in MDA-MB-468 recipient cells incubated for 48 h with 48 h-CM from control, STING$^{-/-}$ or TNFα$^{-/-}$ donor cells treated or not with paclitaxel (upper panel) and in control or IFNAR1$^{-/-}$ MDA-MB-468 recipient cells with 48 h-CM from control donor cells treated or not with paclitaxel (lower panel). **f** Immunoblot and *PMAIP1* qPCR analysis in MDA-MB-468 cells treated for 48 h or not with recombinant TNFα and/or IFNα. **g** Apoptotic effect of recombinant IFNα and/or TNFα on control or NOXA$^{-/-}$ MDA-MB-468 cells treated or not with WEHI-539 during 48 h. Data were collected from at least $n = 3$ independent experiments. Error bars indicate mean $+/-$ SEM; Two-sided paired *t*-test. The symbols correspond to a *p*-value inferior to *0.05 and **0.01.

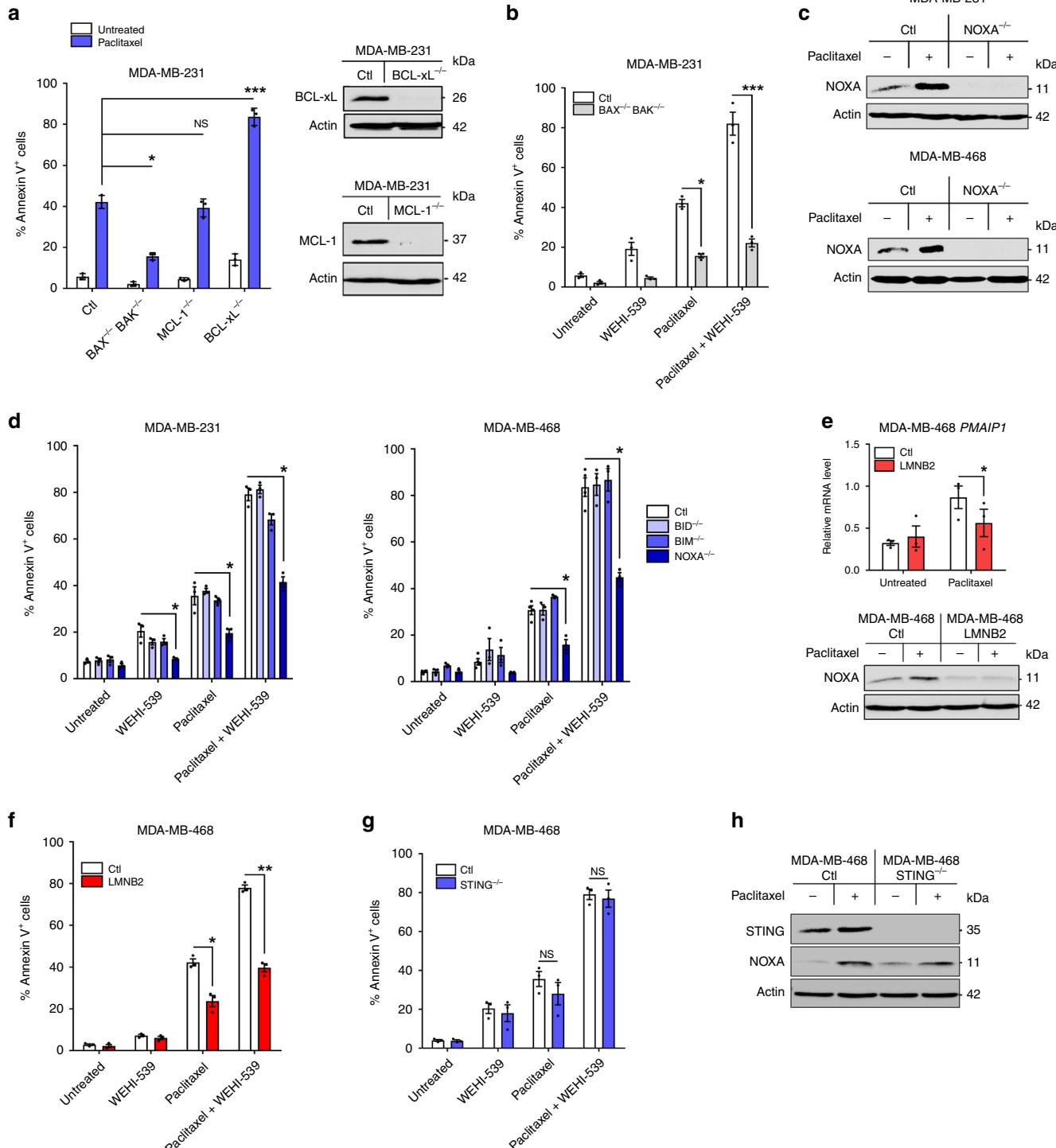

**Fig. 6 NOXA is induced by micronuclei envelope collapse-dependent but STING-independent signals. a** Annexin V assay in control, BCL-xL or MCL-1 KO MDA-MB-231 cells treated or not with paclitaxel for 24 h, washed out, incubated for 48 h (left panel). Analysis of BCL-xL and MCL-1 expression in untreated cells was realized by immunoblot (right panel). **b** Same analysis as in **a** in control or double KO BAX$^{-/-}$ BAK$^{-/-}$ MDA-MB-231 cells treated with paclitaxel for 24 h or not, washed out for 48 h then treated with WEHI-539 for 24 h or not (**c**). Immunoblot analysis in control or NOXA$^{-/-}$ breast cancer cells 48 h after a 24 h-paclitaxel treatment. **d** Same experiment as (**b**) using control, BID$^{-/-}$, BIM$^{-/-}$, or NOXA$^{-/-}$ MDA-MB-231 or MDA-MB-468 cell lines. **e** PMAIP1 qPCR (upper panel) and NOXA immunoblot (lower panel) in control or LMNB2 overexpressing MDA-MB-468 cells 48 h after a 24 h-paclitaxel treatment. **f** Same experiment as (**b**) using control or LMNB2 overexpressing MDA-MB-468 cells. **g** Same experiment as (**b**) using control or STING$^{-/-}$ MDA-MB-468 cells and indicated treatments. **h** NOXA immunoblot analysis in control and STING$^{-/-}$ MDA-MB-468 cells harvested 48 h after a 24 h-paclitaxel treatment. Data were collected from $n = 3$ independent experiments. Error bars indicate mean $+/-$ SEM; Two-sided paired $t$-test. The symbols correspond to a $p$-value inferior to *0.05, **0.01, and ***0.001. NS: not significant.

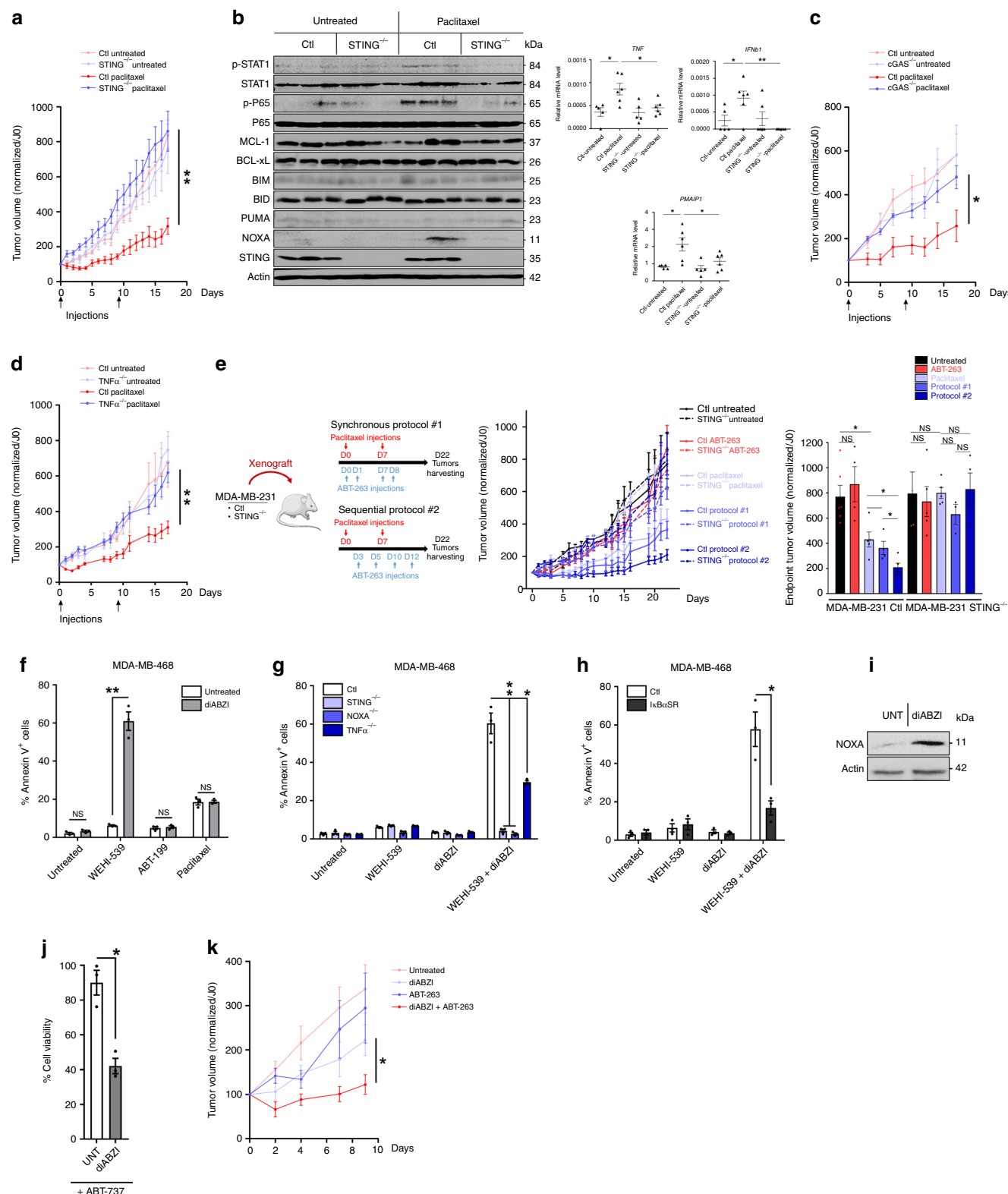

incriminated IRF3, which was established as a key regulator of IFNα/β receptor–mediated feedforward regulation and crosstalk with other pathways[22,36] and detected a slight recruitment of RelA. As cancer cells frequently rely on the complementary activities of BCL-xL and MCL-1 for their survival[15,37,38] we propose that the paracrine effects of paclitaxel, through IFN/TNF dependent NOXA induction, enhance BCL-xL dependency by decreasing the influence of MCL-1. This implies that the paracrine effects of paclitaxel will be overtly lethal provided BCL-xL expression in cancer cells is relatively low, a feature that was associated with sensitivity to chemotherapy in triple negative breast cancers (TNBC)[37]. *MCL1* gene amplification, which was detected in residual TNBC post-neoadjuvant chemotherapy[39] may also counteract these effects.

**Fig. 7 In vivo paclitaxel response relies on STING and is amplified by sequential use of BH3 mimetics. a** Control or STING$^{-/-}$ MDA-MB-231 cell lines were injected in mammary fat pad in immunodeficient mice and when tumors reached about 100 mm$^3$, paclitaxel has been injected twice at D0 and D9 and tumors were calipered every day ($n = 5$ mice per group). **b** Immunoblot (left panel) and qPCR (right panel) analysis of indicated markers in tumors from **a**. **c–d** Same experiment as (**a**) using control and cGAS$^{-/-}$(**c**) or TNFα$^{-/-}$(**d**) MDA-MB-231 cells ($n = 5$ mice per group). **e** Therapeutic protocols using paclitaxel and ABT-263 combinations (left panel) applied to control or STING$^{-/-}$ MDA-MB-231 xenograft model (left panel), tumor volume (mean and SEM) in each mice groups (middle panel) and histogram showing the average tumor size at the end of the experiment (right panel). $n = 6$ mice per group. **f** Annexin V assay in MDA-MB-468 after 48 h treatment with the STING agonist diABZI plus WEHI-539, ABT-199, or paclitaxel or not. **g–h** Same experiment as in **f** using control, STING$^{-/-}$, NOXA$^{-/-}$, or TNFα$^{-/-}$ (**g**) or IκBαSR (NF-κB super repressor) (**h**) MDA-MB-468 cells, treated by diABZI plus WEHI-539 or not. **i** NOXA immunoblot analysis after a 48 h diABZI treatment in MDA-MB-468 cells. **j** Cell viability of PDO after diABZI treatment in presence of ABT-737 or not. **k** Tumor volume in the same xenograft model as in **a** upon sequential treatment with diABZi (2×/week) plus Navitoclax (2×/week) as indicated ($n = 5$ mice per group). Error bars indicate mean $+/-$ SEM; Two-sided unpaired $t$-test. The symbols correspond to a $p$-value inferior to *0.05 and **0.01. NS: not significant.

Importantly, the secretome of chemotherapy-treated cancer cells is double-sided and it may exert protumoral effects in the long term. Chemotherapy triggers IRE1-dependent induction of cytokines to favor breast cancer initiating cell expansion[40] and type I IFN themselves may contribute to this[41]. Chronic activation of type I IFN favors intrinsic resistance to radiotherapy and chemotherapy by inducing specific subsets of Interferon-stimulated genes[42,43] and resistance to immune checkpoint inhibitors by interfering with the PD-1/PD-L1 axis[44]. Likewise, TNF contributes to anti-PD-1-induced TIL cell death[45]. It is thus necessary to fully exploit paclitaxel-induced secreted death signals in a timely manner, as shown here using BH3 mimetics. Numerous preclinical studies, including ours, have put forth the benefits of combining antimitotic treatment with MOMP targeting by BH3 mimetics[46–48]. This study significantly enhances our mechanistic understanding of this combination by establishing a role for proapoptotic intercellular communications, particularly important in vivo. As a consequence, it also brings insight into previous clinical trial where the dual BCL-2/BCL-xL inhibitor ABT-263 or Navitoclax was combined with antimitotic agents[14,49]. Our data clearly establish that BCL-xL is the main antiapoptotic protein antagonizing paracrine effects, advocating for a combination between antimitotic drugs and selective BCL-xL.

The necessity to inhibit BCL-xL raised above, implies that administration schedules need to be strictly defined to exploit antimitotic treatment-induced paracriny despite adverse effects[50,51]. A compromise between the biochemical pathways leading to the secretion of proapoptotic ligands and the own vulnerabilities of cells elaborating them, needs to be found, as BCL-xL inhibition promotes death in cells exposed to the paracrine effects of paclitaxel but also in mitotically blocked cells[7] and in the micronuclei positive subset that produces soluble death signals. The latter cells, in agreement with what was reported in polyploid cells[9,10,52] intrinsically induce NOXA by mechanisms that require further elucidation as they do not require STING, in contrast to T cells[53], but that in all cases renders them highly sensitive to BCL-xL targeting. In vivo, where paracrine effects predominate, the efficiency of combined BCL-xL inhibitor and paclitaxel thus depends on the impact of the BCL-xL inhibition on cells that secrete propaptotic ligands upon addition of paclitaxel. Notably, MOMP per se induces inflammatory secreted signals through downregulation of IAPs and cytosolic release of mitochondrial (mt)DNA[18,54–57]. However, MOMP neither contributes nor amplifies antimitotic stress-induced proapoptotic paracriny, as judged by the lack of effect of BAX/BAK depletion on the induction of extrinsic apoptotic priming by paclitaxel. Instead, BCL-xL depletion, that dramatically enhanced MOMP and subsequent cell death rates, abolished the elaboration of soluble proapoptotic signals in cells exposed to antimitotic therapy. This implies that accelerated caspase-dependent cell death in donor cells would prevent them from supplying extrinsic apoptotic signals and that, paradoxically, short-term caspase

inhibition might be beneficial[54]. The observation that paclitaxel and ABT-263 synergize to prevent tumor growth only when administered sequentially, and not synchronously, is consistent with the notion that donor, paclitaxel sensitive cells should not be killed too fast, and be given enough time to produce a paracrine cytotoxic signal.

It emerges from our study that combining BH3 mimetics and antimitotic treatments is a rational efficient strategy, not because it is less subject to phenotypic variability and biochemical noise than either treatment alone, but because it benefits from these initial cell to cell variations in apoptotic priming (allowing low apoptotically primed cells to produce intercellular death signals) warranting a more homogenous apoptotic response of the population (Supplementary Fig. 7g). Regarding precision medicine, our data underscore that the predictivity of apoptotic priming evaluation in cancer cell populations is enhanced by measuring variability[58] and by assessing dynamic changes during treatments[5]. Regarding clinical management, our data imply that sequences of transient, somehow antagonistic, treatments will best harness tumor therapeutic response and potentially minimize side effects. Temporal sequences of these treatments, initially designed for their intrinsic effects, can be finely tuned by quantifying extracellular signaling molecules related to the cytosolic DNA sensing pathway. Moreover, our demonstration that a recently described STING agonist[21] can influence apoptotic priming by itself reveals that non cell autonomous effects through which differing individuals regulate populational responses to apoptotic stress may be directly manipulated by small molecules.

## Methods

**Cell lines and reagents**. Cells lines were purchased from ATCC and cultured in Dulbecco's Modified Eagle Medium (DMEM) (Gibco, Saint Aubin, France) supplemented with 2 mM glutamine and 1% penicillin/streptomycin (Gibco) and 5% Fetal Bovine Serum (FBS) (Eurobio, Courtaboeuf, France).

Lamin B2 (*LMNB2*) overexpressing cell lines were established by viral infection with retroviruses containing vector coding for *LMNB2* (pQCXIB-mCherry-LMNB2) kindly offered by Lewis C. Cantley's laboratory[59]. Blasticidin was used to select for *LMNB2* expressing cells at 10 μg/ml.

Histone 2B fused to RFP (*H2B-RFP*) expressing cell line was established by viral infection with lentivirus containing vector coding for human *H2B* fused to RFP sequence kindly provided by Jan van Deursen's laboratory[60]. Cells with moderate expression of H2B-RFP were sorted with the BD-FACS ARIA III sorter.

For the CRISPR Cas9-induced knock-out (KO) cell lines, single guide (sg) RNA sequences targeting human genes were designed using the CRISPR design tool (http://crispor.tefor.net). The guide sequences described in Table 1 were cloned in the plentiCRISPRV2 vector that was a gift from Feng Zhang (Addgene plasmid # 52961)[61]. Cells were selected using 1 μg/ml puromycin and KOs were confirmed by immunoblot analysis.

NF-κB super repressor (IκBαSR) expressing cell lines were generated using the pBabe-Puro-IκBα-mutated retroviral plasmid from Addgene #15291 (gift from William Hahn) and puromycin-based selection using 1 μg/ml of the antibiotic.

For the preparation of conditioned media (CM), $1 \times 10^6$ cells were treated as indicated during 24 h, washed 3 times with PBS and cultured in DMEM without FBS for 48 additional hours. CM were then collected, centrifuged (1500 rpm, 5 min) and supplemented with 5% FBS prior to incubation with recipient cells for 48 h with the indicated treatments. For TNFα blocking experiments, neutralizing

antibody (clone D1B4 from Cell Signaling Technology, Danvers, MA, USA) was added in CM (100 ng/ml) 30 min prior to incubation with recipient cells. For thymidine block experiments, thymidine (2 mM) wad added 24 h prior paclitaxel treatment of donor cells.

Cocultures were based on the combined culture of 1:1 mix of breast cancer cells pretreated with paclitaxel during 24 h and untreated H2B-RFP expressing breast cancer cells. After 48 h, cocultures were treated as indicated for additional 48 h.

In vitro treatments were used at the following concentrations: 1 μM Wehi-539 (ApexBio, Houston, TX, USA), 1 μM ABT-737, 1 μM ABT-199 or S63845 (Selleckchem, Houston, TX, USA), 3.5 nM paclitaxel (Sigma-Aldrich, St Quentin Fallavier, France), 10 μM Q-VD-OPh (R&D Systems, Abingdon, UK), 2 μM monensin (Biolegend, London, UK), 10 ng/ml TNFα (Biolegend, London, UK), 2000 UI.μL$^{-1}$ IFNα (Sigma-Aldrich), 2.5 μM Etoposide and 50 nM AZD1152 (Sigma-Aldrich), 1.5 μg/ml cGAMP (Sigma-Aldrich) and 1 μM of diAZBI (Cliniscience).

**Biochemical assays.** For immunoblot analysis, proteins were obtained by lysing cells with CHIP buffer (SDS 1%, EDTA 10 nM, Tris-HCl [pH 8,1] 50 nM plus a cocktail of proteases/phosphatases inhibitors) followed by sonication prior separation on SDS-PAGE and transfer on nitrocellulose membranes. Membranes were then incubated with primary antibodies overnight at 4 °C with the following antibodies used at dilutions recommended by suppliers: Actin (MAB1501) and BIM (AB17003) from Millipore (Molsheim, France); STAT1 (9176), pTyr701-STAT1 (9167), p65 (8242), pSer536-p65 (3033), cGAS (15102), IRF3 (11904), STING (13647), TNFα (3707), cleaved Caspase-3 (9662), PUMA (4976), BID (2002), BAK (3814), and IκBα (4814) from Cell Signaling Technology; BAX (A3533) and BCL-2 (M0887) from Dako (Santa Clara, CA, USA), BCL-xL (ab32370) and NOXA (ab13654) from Abcam (Cambridge, UK); MCL-1 (sc-819) from Santa Cruz Biotechnology (Heidelberg, Germany). Then membranes were incubated with the appropriate secondary antibodies for 1 h at room temperature. Clarity™ western ECL kit (Bio-Rad, Marne la Coquette, France) was used for immunoblot revelation on the ChemiDoc XRS + system (Bio-Rad). The most important blots are supplied uncropped and unprocessed in the Supplementary Figs. 8 and 9 in the Supplementary Information section.

For ELISA assays, levels of IFNα in CM or in tumor slice supernatants were determined according to the manufacturer's protocol (BioLegend, London, UK).

**qPCR analysis.** Total RNA was isolated using Nucleospin RNA plus (Macherey Nagel, Hoerdt, France) and transcribed into cDNA by Maxima First Strand cDNA synthesis Kit (Thermo Scientific, Illkirch, France). Quantitative RT-PCR (qPCR) was performed using the EurobioGreen qPCR Mix Lo-Rox with qTOWER (Analityk-jena, Jena, Germany). Reaction was done in 10 μl final with 4 ng RNA equivalent of cDNA and 150 nM primers. Primers sequences used for DNA amplification are listed in Table 2.

**Flow cytometry analysis.** Apoptosis analysis was evaluated by staining cells with Annexin V-FITC (Miltenyi Biotec, Bergisch Gladbach, Germany) or Annexin-V-APC (BD Pharmingen, le Pont de Claix, France) for coculture experiments according to manufacturer's instructions. For cell cycle analysis, cells were fixed with ethanol 70% for 1 h, and stained with propidium iodide (10 μg.ml$^{-1}$). Flow cytometry analysis was performed on FACS Accuri C6 plus (BD Biosciences). All experiments were repeated at least 3 times.

**Immunofluorescence and immunohistochemitry.** For cGAS immunofluorescence staining, cells were grown on glass coverslips, treated as indicated and fixed in 4% paraformaldehyde (PFA) in PBS for 20 min at room temperature. Cells were permeabilized in 0.5% Triton X-100 for 5 min prior to blocking for 30 min with 3% BSA in PBS. Coverslips were then incubated with cGAS antibody (clone D1D3G, Cell Signaling Technology, 1:200) for 1 h, followed by incubation with a secondary antibody for 45 min. Coverslips were mounted using ProLong Diamond Antifade Mountant with DAPI (Invitrogen, Carlsbad, CA, USA) and imaged using a Zeiss Axioplan II fluorescence microscope. Micronuclei positive cells were counted manually. For PDX tumor analysis by immunohistochemistry (IHC), 3 μm-thick tissue sections were dried at 37 °C overnight, deparaffinized, and pretreated at 95 °C for antigen retrieval in a basic buffer (CC1, Cell Conditioning Medium-1, pH = 8.4, Ventana Medical Systems, Tucson, AZ) in a BenchMark XT immunostainer (Ventana Medical Systems). Sections were stained at 37 °C with anti-ER rabbit monoclonal antibody (clone SP1, Abcam), anti-PR rabbit monoclonal antibody (clone 1E2, Ventana Medical Systems), or anti-cGAS rabbit monoclonal antibody (clone D1D3G, Cell Signaling Technology). Chromogenic detection was performed using the Ventana iView DAB IHC detection kit, with replacement of the secondary antibody by a polyclonal goat anti-rabbit immunoglobulin secondary antibody from Dako (reference E0433), followed by counterstaining with hematoxylin-II and bluing reagent. Negative controls were obtained by replacement of the primary antibodies with normal rabbit serum (Negative Control Rabbit Ig, 10 μg/ml, Ventana Medical Systems). The number of cGAS-positive micronuclei among at least 500 neoplastic cells was calculated by manual image analysis using the image J software (National Institute of Health, Research Service Branch, Bethesda, Maryland, USA).

**Preclinical breast cancer ex vivo assay.** Fresh human breast cancer samples from patients were collected after surgical resection at the Institut de Cancerologie de l'Ouest, Rene Gauducheau, Nantes, France, between 2009 and 2017. As required by the French Committee for the Protection of Human Subjects, informed consent was obtained from patients and the local ethic committee approved protocols (2012-A00682-41). The tumors were cut into thin slices (250 μm) by using a

---

### Table 1 Guide sequences used in CRISPR-Cas9-based generation of KO cell lines.

| | |
|---|---|
| Human *BAX* | AGTAGAAAAGGGCGACAACC |
| Human *BAK* | GCCATGCTGGTAGACGTGTA |
| Human *PMAIP1* | TCGAGTGTGCTACTCAACTC |
| Human *BID* | CACCGTCAACAACGGTTCCAGCCTC |
| Human *BCL2L11* | CACCGAGTTCTGAGTGTGACCGAGA |
| Human *BCL2L1* | GCAGACAGCCCCGCGGTGAA |
| Human *MCL1* | CGCGGTGACGTCGGGGACCT |
| Human *STING* | GCAGGCACTCAGCAGAACCA |
| Human *TNF* | TGAAAGCATGATCCGGGACG |
| Human *IFNAR1* | GCGGCTGCGGACAACACCCA |
| Human *cGAS* | CACCGAGACTCGGTGGGATCCATCG |

---

### Table 2 Primer sequences used in RTqPCR.

| | |
|---|---|
| *RPS18* | 5′-ATCCCTGAAAAGTTCCAGCA/CCCTCTTGGTGAGGTCAATG-3′ |
| *RPLP0* | 5′-AACCCAGCTCTGGAGAAACT/CCCCTGGAGATTTTAGTGGT-3′ |
| *HPRT1* | 5′-ATGCTGAGGATTTGGAAAGG/GATGTAATCCAGCAGGTCAGC-3′ |
| *PMAI1* | 5′-CTCTGTAGCTGAGTGGGCG/CGGAAGTTCAGTTTGTCTCCA-3′ |
| *BBC3* | 5′-ACCTCAACGCACAGTACGA/GCACCTAATTGGGCTCCATC-3′ |
| *BID* | 5′-GAAGCGGGTAGTCGACCG/GGAACCGTTGTTGACCTCAC-3′ |
| *BCL2L11* | 5′-GCCTTCAACCACTATCTCAG/TAAGCGTTAAACTCGTCTCC-3′ |
| *BCL2L1* | 5′-TTCAGTGACCTGACATCCCA/TCCACAAAAGTATCCCAGCC-3′ |
| *MCL1* | 5′-TCGGTACCTTCGGGAGCAGGC/CCCAGTTTGTTACGCCGTCGCT-3′ |
| *BCL2* | 5′-TCTTCAGAGACAGCCAGGAG/CCTTCTTTGAGTTCGGTGGG-3′ |
| *TNF* | 5′-CTGCACTTTGGAGTGATCGG/CTCGGGGTTCGAGAAGATGA-3′ |
| *IFNβ* | 5′-ATGACCAACAAGTGTCTCCTCC/GCTCATGGAAAGAGCTGTAGTG-3′ |
| *IFIT1* | 5′-TTGATGACGATGAAATGCCTGA/CAGGTCACCAGACTCCTCAC-3′ |
| *TMEM173* | 5′-CCTGAGTCTCAGAACAACTGCC/GGTCTTCAAGCTGCCCACAGTA-3′ |
| *OASL* | Hs_OASL_va.1_SG QuantiTect Primer Assay QT01011451 from Qiagen |
| *MX1* | Hs_MX1_1_SG QuantiTect Primer Assay QT00090895 from Qiagen |
| *BIRC2* | 5′-GAATCTGGTTTCAGCTAGTCTGG/GGTGGGAGATAATGAATGTGCAA-3′ |
| *BIRC3* | 5′-AAGCTACCTCTCAGCCTACTTT/ CCACGGCAGCATTAATCACAGGA-3′ |

vibratome (Microm International, ThermoFischer Scientific, Ilikirch, France) and incubated for 48 h with 700 nM paclitaxel (Sigma-Aldrich), 1 μM ABT-737 (Selleckchem) or without treatment. Tumor slices were then fixed in 10% buffered formalin and paraffin embedded before IHC analysis to assess tumoral cell apoptosis with active caspase-3 antibody (clone D64E10 Ozyme, St Quentin en Yvelines, France) and proliferation with Ki-67 antibody (clone 30-9, Roche Diagnostics, Meylan, France), as described in ref. [20]. Data are represented as the Δ percentages of positive tumoral cells in treated conditions minus positive cells in untreated conditions.

**Patient-derived organoid cultures**. Breast cancer tissues from patients that underwent surgical tumor resection in the Institut de Cancerologie de l'Ouest, Rene Gauducheau, Nantes, France, after informed consent, or tumors harvested in PDX, were processed through a combination of mechanical disruption and enzymatic digestion to generate patient-derived organoids (PDO) or PDX-derived organoids (PDX organoids) as decribed by Sachs et al.[62]. Briefly, isolated cells were plated in adherent basement membrane extract drops (Cultrex Pathclear Reduced Growth Factor BME (Biotechne), and overlaid with optimized breast cancer organoid culture medium. Medium was changed every 4 days and organoids were passaged every 1–4 weeks. For cell viability assay, organoids were split, strained <70 μm, and allowed to grow in BME in white 96-well plates for 3 days before addition of drugs in triplicate (same concentrations as for breast cancer cell lines) in synchronous or sequential treatment as indicated, ATP was measured using the CellTiter-Glo 3D Reagent (Promega) after 4 days, following supplier recommendations and the lumino/fluorometer FLUOstar Omega (BMG Labtech, Ortenberg, Germany). Data were analyzed using GraphPad Prism 6 after normalization by the reference treatment.

**In vivo experiments**. Animal experiments were performed in accordance with the French regulations and approved by the local animal ethics committee (APAFIS#9634-201704191725600 and APAFIS#5114-2016042011155376). PDX models were generated by transplanting freshly obtained surgically excised breast tumor specimens from patients in the mammary fat pad of pre-pubescent female NOD-scid IL2Rgamma$^{null}$ (NSG) supplemented with β-estradiol in the drinking water and subcutaneous pellet as described in ref. [63]. PDX#248 and #249 were established from ERα + /PR + /non amplified-HER2 primary breast tumors from untreated patients and PDX#306 from an ERα-/PR-/HER2na tumor from a patient who benefited from neoadjuvant treatment. ERα and PR primary tumor expression by IHC and TP53 mutational status defined by NGS were maintained in PDX tumors. In orthotopic xenograft experiments with parental (control), STING$^{-/-}$, CGAS$^{-/-}$, or TNF$^{-/-}$ MDA-MB-231 cell lines, 60 μl of mix (1:1) PBS-low enrichment Matrigel (BD Biosciences) containing $4 \times 10^6$ cells were injected in the mammary fat-pad of pre-pubescent female NOD-scid IL2Rgamma$^{null}$ (NSG). When mice tumors became palpable, they were calipered twice per week to monitor growth kinetics. Mice were treated as indicated when tumor size reached 80–150 mm$^3$. Paclitaxel (Sigma-Aldrich), and ABT-263 (Selleckchem) were injected intraperitonealy and used at the following concentrations of 10 mg/kg and of 100 mg/kg, respectively. The diABZI STING agonist-1 (Cliniscienses) was injected intravenously at 1.5 mg/kg. Tumor volumes were callipered daily and quantified using the formula $4/3\pi(\sqrt{dxD})/2)^3$ and normalized by tumor volume measured just before beginning treatment. Mice were killed humanely when tumors reached clinical endpoints.

**Gene expression profiling**. Gene expression analysis of untreated human breast tumors was performed using Affymetrix® Human Genome U133 Plus 2.0 Arrays (Affymetrix®, Santa Clara, CA, USA) measuring over 54,000 transcripts representing over 20,000 human genes as described in ref. [64]. cRNA synthesis and labeling, as well as chip hybridization, washing, and image scanning were performed according to the manufacturer's protocol. All microarrays complied with quality criteria. The Affymetrix® CEL files (raw data) were MAS5-normalized in the Affymetrix Expression Console (v1.3.1) and then log2-transformed. Twelve gene expression signatures (GES) were selected for evaluation of different biological pathways and features. Five GES were for immune response dissection (interleukin-8 [IL-8], MHC-1, MHC-2, STAT1, and type I interferon [type I IFN]), 2 for metabolism evaluation (glycolysis and iron [iron regulatory gene signature: IRGS]), and 5 for critical biological pathways in cancer (chromosomal instability [CIN], mitochondrial oxidative phosphorylation [MITO/OXPHOS], proliferation, reactive stroma and wound response). GES scores were calculated for each patient using average expression or weighted average expression of combinations of gene or probe expressions (Supplementary methods). Stromal and immune scores were computed from the Affymetrix gene expression analysis using Estimate R package (http://bioinformatics.mdanderson.org/estimate/rpackage.html)[65].

For PDX gene expression, 3′ digital gene expression (DGE) RNAseq was performed according to Kilens et al.[66] Briefly, the libraries were prepared from 10 ng of total RNA. The mRNA poly(A) tails were tagged with universal adapters, well-specific barcodes and unique molecular identifiers (UMIs) during template-switching reverse transcriptase. Barcoded cDNAs from multiple samples were then pooled, amplified and tagmented using a transposon-fragmentation approach which enriches for 3′ends of cDNA. A library of 350–800 bp was run on an

Illumina HiSeq 2500 using a Hiseq Rapid SBS Kit v2–50 cycles (ref FC-402–4022) and a Hiseq Rapid PE Cluster Kit v2 (ref PE-402-4002). DGE profiles were generated by counting for each sample the number of unique UMIs associated with each RefSeq genes. DESeq 2 was used to normalize expression with the DESeq function. Differentially expressed gene sets were further evaluated using enrichment analysis tools from Enrichr website (http://amp.pharm.mssm.edu/Enrichr)[67]. Three mice per group for PDX #249, 2 for PDX #248, and PDX #306, were analyzed by DGE-RNAseq.

**Statistical analysis**. Student's t test was used for statistical analysis with GraphPad Prism 5.0 Software. Errors bars represent standard errors of mean (SEM). The symbols correspond to a P-value inferior to *0.05, **0.01, ***0.001.

**Reporting summary**. Further information on research design is available in the Nature Research Reporting Summary linked to this article.

## Data availability

All the other data supporting the findings of this study are available within the article and its supplementary information files and from the corresponding author upon reasonable request. The source microarray data underlying Fig. 2 and supplementary 2 have been deposited in the NCBI Gene Expression Omnibus database under the accession code GSE140489. The DGE-RNAseq count table corresponding to Fig. 3b is provided in Mendeley Data (https://doi.org/10.17632/bsjc7f3hck.1).

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

## Acknowledgements

We thank the Ligue contre le Cancer Grand Ouest for supporting S.L. fellowship. We thank members of the "Stress adaptation and tumor escape" laboratory for their support and Julie Roul for her technical support in qPCR analysis. We are grateful to Victor Simmet and Alain Morel for NGS analysis, to Jonathan Lopez and to Samuel Bakhoum for providing us CRISPR-Cas9 material and *LMNB2* plasmid, respectively. We thank the CytoCell, MicroPiCell, GenoCellEdit, and UTE facilities (SFR-Santé F Bonamy UMS 016) for technical support and LabEx IGO project (ANR-11-LABX-0016-01) for providing mice. This work was supported by the Ligue contre le Cancer (44, 22, 53, 56, and 85), the Canceropole Grand Ouest, the Region Pays de la Loire (MATURE project 2017–18) and the Agence Nationale de la Recherche (Grands defis societaux 2015–2020, Antinetrex project). This paper was prepared in the context of the SIRIC ILIAD program supported by the French National Cancer Institute national (INCa), the Ministry of Health and the Institute for Health and Medical Research (Inserm) (SIRIC ILIAD, INCa-DGOS Inserm-12558).

## Author contributions

S.L., P.P.J., and S.B.N. conceived the study and designed the work. S.L., N.B., A.F., C.V., A.D., M.F., L.M., S.B.N. performed experiments. S.L., F.G., L.D., A.M.A., H.L., P.J., P.P.J., and S.B.N. contributed to data analysis. M.C., F.N., O.K., and D.L. provided tumor samples and histological analysis. P.P.J. and S.B.N. obtained funding and wrote the paper.

## Competing interests

The authors declare no competing interests.

**Additional information**

