## [Peer Review File · Nature Communications]

Reviewers' comments:

Reviewer #1 (Remarks to the Author): Expert in breast cancer cell death

Lohard et al. report that paclitaxel causes the generation of micronuclei, which in resistant breast cancers cells, triggers cGAS/STING-mediated production of TNF and type-I interferon. This in turn, in a auto- and paracrine fashion, results in NOXA expression, driving MOMP and cell death. They demonstrate that STING-mediated induction of NOXA is required for paclitaxel responses in vivo. What is more, they demonstrate that sequential, but not simultaneous, administration of BH3 mimetics synergises with paclitaxel in killing cancer cells.

All in all this is an interesting manuscript that reports an important and novel finding. It suggests that treatment with anti-mitotic agents causes STING-dependent cell-cell communication that is driven by resistant cells and that propagates to sensitise cells in a auto-and paracrine fashion, killing cells via up-regulation of NOXA. It also suggests that BH3 mimetics should not be simultaneously applied with mitotic poisons as this sensitises resistant cells to cell death, nullifying STING activation and subsequent cell-cell communication, therefore weakening the paracrine effect of paclitaxel.

While the ms is interesting, there are several aspects that should be addressed. Particularly, I strongly suggest to alter the flow of the data presented. In its current format the ms is difficult to read and flows poorly. A better approach would be: Paclitaxel causes the generation of micronuclei -> this activates cGAS/STING -> this results in TNF and IFN production -> these cytokines drive NOXA-dependent apoptosis in an autocrine and paracrine fashion -> this priming makes them sensitive to BH3 mimetics -> and this new knowledge can be exploited to improve anti-cancer strategies in vivo. The authors should be advised to rethink the structure of the paper.

Other issues:

1) The authors should better highlight previously known observations. This does not diminish their findings but puts them into a better context. For example, the paclitaxel/BH3 mimetics combination, Navitoclax/paclitaxel synergy was observed in several early 2010s publications [4-6] and in more recent publications [7, 8]. The role of BCL-XL and MCL1 downregulation in the response was described [4, 5, 8], as well as the one of TNF/NF-kB [7]. A phase I clinical trial was run [9].

Therefore, this study should better highlight the novel aspect of their finding, which is clearly there, but which is somewhat hidden. Particularly, they should focus on offering a better characterisation of the 'auto- and paracrine' relationship and on the new treatment regimen offered in Figure 6.

2) Figure 2, it would be beneficial if the authors could demonstrate that cGAS/STING mediates upregulation of TNF and type I interferon. Can they use CRISPR/Cas9 to generate cGAS or STING KO PDXs, or patient-derived organoids from these lines. Currently it is possible to grow PDXs short term in culture (see Sachs et al from the Clevers lab). While this would be good to have it is not essential as this is technically challenging. But it would be good to have because it is well known that paclitaxel triggers activation of NFk-B and interferon, but the signalling pathway is not necessarily clear.

3) Figure 3 assesses the ability of conditioned medium from paclitaxel treated cells to prime recipient cells to cell death. The sensitization to the WEHI-539 compound is not particularly strong. The fold difference between Untreated and paclitaxel-treated conditioned medium seems to be about the same. The overall amplitude seems to be different following WEHI-539 treatment. As this is a snapshot (48h), a longer term assay (or time course experiment) would be desirable to corroborate their conclusion. Also conditioned media of cells that were stressed in other ways (ways that do not activate the cGAS/STING pathway) will be required. How much cell death occurs

in 'donor cells', and what is the cell death modality of such cells? Does the death of donor cells contribute to this phenomenon?

Serum deprivation does arrest cells in G1, but does a lot more to cells than simply blocking their proliferation. Hence, it is not entirely clear whether the transcriptional/translational capacity of such cells (following serum starvation) affects the conclusion. Cells should be arrested in other ways too.

The effect of knocking out STING or IFNAR1 is modest, even though the authors describe it as 'abrogate' the sensitivity. 'Abrogate' is clearly the wrong term here. What is the effect of IFNAR1/TNFR1 double knockout cell lines? Also, it might be worthwhile to repress NF-kB (on its own and in combination with loss of STING) via expression of the NF-kB super-repressor in 'doner' cells to evaluate contribution of NF-kB versus interferon. It is well known that paclitaxel potentially triggers NF-kB activation.

4) How does the combination of INF α and TNF drive NOXA accumulation?

5) Figure 6: I am sure that they authors will be aware of the phase I clinical trial with Paclitaxel/Navitoclax in solid tumors that was conducted and prematurely terminated in 2012. The results were disappointing due to significant hematological and non-hematological toxicity [9]. No other phase I trial has been run with this combination since. The protocol used was similar to Protocol #1 (Fig. 6). Protocol #2 similarly uses BH3 mimetics with a toxicity profile that is likely to be similar to the one of the clinical trial. Given the recent data on Venetoclax, it might be worthwhile to explore this BH3 mimetic in combination with paclitaxel, as this is trialled at present in breast cancer [10, 11].

It is difficult to conclude that Protocol 1 is worse than protocol 2. Obviously Navitoclax has a particular in vivo half-life. In protocol 2 it seems that the authors administered Navitoclax in a way that maintains a certain level of this drug throughout the treatment, while in protocol one there is a Navitoclax drug holiday. Without a proper characterisation of the kinetics of TNF/IFN induction and NOXA upregulation in vivo, it is difficult to make a strong conclusion.

1. Harding, S.M., Benci, J.L., Irianto, J., Discher, D.E., Minn, A.J., and Greenberg, R.A. (2017). Mitotic progression following DNA damage enables pattern recognition within micronuclei. *Nature* 548, 466-470.
2. Mackenzie, K.J., Carroll, P., Martin, C.A., Murina, O., Fluteau, A., Simpson, D.J., Olova, N., Sutcliffe, H., Rainger, J.K., Leitch, A., et al. (2017). cGAS surveillance of micronuclei links genome instability to innate immunity. *Nature* 548, 461-465.
3. Pineda, J.J., Miller, M.A., Song, Y., Kuhn, H., Mikula, H., Tallapragada, N., Weissleder, R., and Mitchison, T.J. (2018). Site occupancy calibration of taxane pharmacology in live cells and tissues. *Proc Natl Acad Sci U S A* 115, E11406-E11414.
4. Shi, J., Zhou, Y., Huang, H.C., and Mitchison, T.J. (2011). Navitoclax (ABT-263) accelerates apoptosis during drug-induced mitotic arrest by antagonizing Bcl-xL. *Cancer Res* 71, 4518-4526.
5. Tan, N., Malek, M., Zha, J., Yue, P., Kassees, R., Berry, L., Fairbrother, W.J., Sampath, D., and Belmont, L.D. (2011). Navitoclax enhances the efficacy of taxanes in non-small cell lung cancer models. *Clin Cancer Res* 17, 1394-1404.
6. Wong, M., Tan, N., Zha, J., Peale, F.V., Yue, P., Fairbrother, W.J., and Belmont, L.D. (2012). Navitoclax (ABT-263) reduces Bcl-x(L)-mediated chemoresistance in ovarian cancer models. *Mol Cancer Ther* 11, 1026-1035.
7. Panayotopoulou, E.G., Muller, A.K., Borries, M., Busch, H., Hu, G., and Lev, S. (2017).

Targeting of apoptotic pathways by SMAC or BH3 mimetics distinctly sensitizes paclitaxel-resistant triple negative breast cancer cells. *Oncotarget* 8, 45088-45104.

8. Vallet, S., Fan, F., Malvestiti, S., Pecherstorfer, M., Sattler, M., Schneeweiss, A., Schulze-Bergkamen, H., Opferman, J.T., Cardone, M.H., Jager, D., et al. (2019). Rationally derived drug combinations with the novel Mcl-1 inhibitor EU-5346 in breast cancer. *Breast Cancer Res Treat* 173, 585-596.

9. Vlahovic, G., Karantza, V., Wang, D., Cosgrove, D., Rudersdorf, N., Yang, J., Xiong, H., Busman, T., and Mabry, M. (2014). A phase I safety and pharmacokinetic study of ABT-263 in combination with carboplatin/paclitaxel in the treatment of patients with solid tumors. *Invest New Drugs* 32, 976-984.

10. Vaillant, F., Merino, D., Lee, L., Breslin, K., Pal, B., Ritchie, M.E., Smyth, G.K., Christie, M., Phillipson, L.J., Burns, C.J., et al. (2013). Targeting BCL-2 with the BH3 mimetic ABT-199 in estrogen receptor-positive breast cancer. *Cancer Cell* 24, 120-129.

11. Lok, S.W., Whittle, J.R., Vaillant, F., Teh, C.E., Lo, L.L., Policheni, A.N., Bergin, A.R.T., Desai, J., Ftouni, S., Gandolfo, L.C., et al. (2019). A Phase Ib Dose-Escalation and Expansion Study of the BCL2 Inhibitor Venetoclax Combined with Tamoxifen in ER and BCL2-Positive Metastatic Breast Cancer. *Cancer Discov* 9, 354-369.

12. Gandhi, L., Camidge, D.R., Ribeiro de Oliveira, M., Bonomi, P., Gandara, D., Khaira, D., Hann, C.L., McKeegan, E.M., Litvinovich, E., Hemken, P.M., et al. (2011). Phase I study of Navitoclax (ABT-263), a novel Bcl-2 family inhibitor, in patients with small-cell lung cancer and other solid tumors. *J Clin Oncol* 29, 909-916.

Reviewer #2 (Remarks to the Author): Expert in STING signalling

STING-dependent paracrine shapes apoptotic priming of breast tumors by anti-mitotic treatment
Here, Lohard et al. investigated the mechanisms of action of the chemotherapeutic drug Paclitaxel (PTX) in breast cancer cells, as well as the potential of combinatorial treatment approaches involving PTX and BH3 mimetics for treatment of breast cancer. Interestingly, they found that STING signaling within the PTX-hit cycling tumor cells aids in apoptotic priming of the neighbouring tumor cells, which are not sensitive to PTX treatment, via the mechanisms regulated by type I IFNs and TNF α produced by PTX-hit tumor cells in a STING-dependent manner. On the other hand, they showed that BH3 mimetics treatment following PTX treatment showed significantly improved anti-tumor effect compared to PTX singular treatment or simultaneous PTX and BH3 mimetics treatment strategies in a STING-dependent manner, emphasizing in vivo relevance of their in vitro data using breast cancer cells from human patients. Therefore, these data demonstrated that such combinatorial therapies might be advantageous compared to singular PTX or BH3 mimetics treatment options only if the treatment protocol is carefully optimized.

Major Comments:

1- Here, authors used a combination therapy involving PTX and BH3 mimetics to improve anti-tumor effect of PTX in a murine breast cancer model. Since both PTX and anti-apoptotic therapy could have some serious side effect due to targeting of the cells other than cancer cells in the body, doesn't the authors think that it would be better to combine STING agonists with PTX instead of BH3 mimetics? What do you think are the advantages of PTX + BH3 mimetics combination therapy over PTX + STING agonist combination therapy?

2- In this study, authors are mainly using annexin V staining for measuring apoptotic cell death. However, as other types of cell death, such as necrosis, can also be detected by annexin V staining, it is better to include one more definitive staining, such as propidium iodide staining, which can discriminate between apoptotic and necrotic cells. Regarding this information, do the authors think that PTX induces tumor cell death via apoptosis but not via necrosis or other types of cell death in their experiments?

3- In figure 2j and supplementary figure 2f, IFN α production in the untreated STING KO MDA-MB-468 cells is higher compared to WT cells. Is this a significant increase and do the authors have any explanation for this? Is this due to the method used for preparation of these STING KO cells (CRISPR-Cas9 method)? If so, do you think that it would affect the interpretation of the current data?

4- Do the authors think that this STING-dependent paracrine mechanisms of action of PTX revealed in breast cancer cells apply for other types of cancer cells, for which PTX therapy is being used in the clinic (e.g. ovarian or non-small cell lung cancer)? In other means, is this STING-dependent paracrine mechanism a general mechanism by which anti-mitotic drugs work in the cancer cells? Have the authors tested other tumor cell types or anti-mitotic drugs for answering this question?

5- In this study, authors show that PTX acts on tumor cells to activate STING pathway most likely via mitotic arrest-dependent micronuclei formation and subsequent leakage of those micronuclei-bound nucleic acids into the cytosol to activate cGAS/STING pathway. Regarding those nucleic acids and their sensors, do the authors know whether those nucleic acids are composed of DNA-only (or RNA or DNA/RNA hybrids)? Also, is cGAS the only cytosolic nucleic acid sensor responsible for detection of these nucleic acids (what about AIM2 or others) in these experimental settings?

Minor Comments:

1- English editing is recommended for this manuscript, especially for discussion section. It is difficult to understand the meaning of the sentences.

Reviewer #3 (Remarks to the Author): Expert in DNA damage

This manuscript describes the role of paclitaxel-induced extrinsic as well as intrinsic signals in priming breast cancer cells to BH3 mimetic -induced cell death. The study is very well-conducted uses human samples manipulated ex vivo, PDXs and genetically engineered cell line models. Generally, the data presented is very convincing and manuscript proposes a novel and interesting mechanism that impinges on modulating sensitivity to paclitaxel to improve patient outcome. In my opinion, the study does achieve high-level of novelty and conceptual impact one expects of papers in Nat Commun.

Rebuttal letter

Reviewer #1

All in all this is an interesting manuscript that reports an important and novel finding. It suggests that treatment with anti-mitotic agents causes STING-dependent cell-cell communication that is driven by resistant cells and that propagates to sensitise cells in an auto- and paracrine fashion, killing cells via up-regulation of NOXA. It also suggests that BH3 mimetics should not be simultaneously applied with mitotic poisons as this sensitises resistant cells to cell death, nullifying STING activation and subsequent cell-cell communication, therefore weakening the paracrine effect of paclitaxel. While the ms is interesting, there are several aspects that should be addressed. Particularly, I strongly suggest to alter the flow of the data presented. In its current format the ms is difficult to read and flows poorly. A better approach would be:

Paclitaxel causes the generation of micronuclei -> this activates cGAS/STING -> this results in TNF and IFN production -> these cytokines drive NOXA-dependent apoptosis in an autocrine and paracrine fashion -> this priming makes them sensitive to BH3 mimetics -> and this new knowledge can be exploited to improve anti-cancer strategies *in vivo*. The authors should be advised to rethink the structure of the paper.

We thank this reviewer for his positive comments about the importance and the novelty of our work. We followed his suggestion to change the outline of the manuscript. In this revised version, we directly start by the description of the effect of paclitaxel on cGAS/STING pathway activation through micronuclei generation, accompanied by the elaboration of a proapoptotic secretome. We then describe the wide transcriptome analysis of *ex vivo* and PDX models that hint on type I IFN and TNF as soluble contributors to the pro-apoptotic effects of paclitaxel and we follow by functional experiments that incriminate these cytokines in cGAS/STING dependent paracrine induced by paclitaxel, inducing NOXA in recipient cells. In the last section, we mechanistically discriminate cell autonomous and non cell autonomous effects of paclitaxel by showing that only the latter requires STING expression, show that non cell autonomous effects produced by viable cells are critical *in vivo* and follow by the comparison between synchronous and sequential protocols in paclitaxel/Navitoclax *in vivo* administration. This part is now incremented with data using the new STING agonist (diABZI) recently published by Ramanjulu JM *et al* in Nature Dec 2018 and now commercially available that we show to efficiently combine with Navitoclax.

1) The authors should better highlight previously known observations. This does not diminish their findings but puts them into a better context. For example, the paclitaxel/BH3 mimetics combination, Navitoclax/paclitaxel synergy was observed in several early 2010s publications [4-6] and in more recent publications [7, 8]. The role of BCL-XL and MCL1 downregulation in the response was described [4, 5, 8], as well as the one of TNF/NF- κ B [7]. A phase I clinical trial was run [9]. Therefore, this study should better highlight the novel aspect of their finding, which is clearly there, but which is somewhat hidden. Particularly, they should focus on offering a better characterisation of the 'auto- and paracrine' relationship and on the new treatment regimen offered in Figure 6.

To better underscore our novelty in the context of paclitaxel/BH3 mimetic combination and preceding observations, we have included more publications proposed by the reviewer, and 2 additional ones, in a new part of the discussion section.

2) Figure 2, it would be beneficial if the authors could demonstrate that cGAS/STING mediates upregulation of TNF and type I interferon. Can they use CRISPR/Cas9 to generate cGAS or STING KO PDXs, or patient-derived organoids from these lines. Currently it is possible to grow PDXs short term in culture (see Sachs *et al* from the Clevers lab). While this would be good to have it is not essential as this is technically challenging. But it would be good to have because it is well

known that paclitaxel triggers activation of NFk-B and interferon, but the signalling pathway is not necessarily clear.

During the revision process, we implemented *ex vivo* cultures of breast cancer organoids derived from PDX or from fresh surgical specimen following the protocol published by Sachs *et al* (Cell 2018 Jan 11;172(1-2):373-386.e10). However, we have not yet succeeded in producing CRISPR-mediated KO in these models, which may be due to inefficient antibiotic-based cell selection. If we could not, therefore, formally incriminate STING in the inflammatory effects of *in vivo* paclitaxel treatment of PDX, we nevertheless confirmed that *ex vivo* treatment of PDX-derived organoids with a STING agonist phenocopied paclitaxel ability to induce *TNF* and *IFNB1* (Figure 4e). As discussed below (see reviewer #2), the STING inhibitor H-151 described in Haag SM *et al* (Targeting STING with covalent small-molecule inhibitors, *Nature* 559,269 (2018)) was tested but this molecule appeared to be toxic for breast cancer cells even at the low dose, in a STING-independent way since even STING KO cells were impacted. We therefore stopped experiments using this compound. Moreover, our revised manuscript shows new experiments using CRISPR-based cGAS KO in breast cancer cell lines that confirm the major role of cGAS in the induction of a proapoptotic secretome by paclitaxel (and the induction of type I IFN and TNF in particular) and, accordingly, in the *in vivo* tumor response to this antimetabolic (Figures 1b, 4c and 7c). Our data provide further evidence that cGAS/STING activation contributes to paclitaxel induction of an inflammatory phenotype.

3) Figure 3 assesses the ability of conditioned medium from paclitaxel treated cells to prime recipient cells to cell death. The sensitization to the WEHI-539 compound is not particularly strong. The fold difference between Untreated and paclitaxel-treated conditioned medium seems to be about the same. The overall amplitude seems to be different following WEHI-539 treatment. As this is a snapshot (48h), a longer term assay (or time course experiment) would be desirable to corroborate their conclusion.

Regarding, MDA-MB-468 cell line shown in the Figure 1g, media conditioned by paclitaxel-treated cells induced an increase in cell death rates above control of 5% in the absence of the BH3 mimetic WEHI-539 and of 35% in presence of the BH3 mimetic after a 48h-treatment. We performed longer term assays as asked by this reviewer: we have now included clonogenic assays following exposure to conditioned media alone or in addition to BH3 mimetics and observed that the proapoptotic paracrine effect synergized with WEHI-539 to decrease cell colony formation. This is shown in the new Supplementary Fig 1e.

We also performed time course experiments as suggested. These indicated that the higher apoptotic effect of WEHI-539 combined with conditioned media from paclitaxel-treated cells was obtained at 48h and lasted at least 96h while the effects of conditioned media alone remained minimal (compare pink and red bars in Rebuttal Figure 1 below).

Rebuttal Figure 1 : Time course analysis of cell death induced by CM from paclitaxel-treated or untreated MDA-MB-468 cells in presence of the BCL-xL inhibitor WEHI-539 or not.

Also conditioned media of cells that were stressed in other ways (ways that do not activate the cGAS/STING pathway) will be required.

As suggested, we have tested other drugs and evidenced that the proapoptotic paracrine effect coincided with mitotic perturbation since the Aurora-B kinase inhibitor phenocopied paclitaxel-induced paracrine effects (included in the new Supplementary Fig 1o) in contrast to the genotoxic drug etoposide. Media conditioned by cells treated with the latter compound, indeed, did not enhance apoptotic priming in recipient cancer cells even though this compound could directly kill cancer cells when combined with WEHI-539 (data included in the new Supplementary Fig 1p).

How much cell death occurs in ‘donor cells’, and what is the cell death modality of such cells? Does the death of donor cells contribute to this phenomenon?

Cell death triggered by paclitaxel appears to follow an apoptotic mode since genetic deletion of BAX and BAK by double CRIPR-based KO considerably alleviates its efficacy (as shown in Fig 6a in MDA-MB-231 cells and Rebuttal figure 2 below in MDA-MB-468 cells where apoptotic rate decreases from 25% to 5%) and since Q-VD-OPh abrogated cell death rates (included in the new Supplementary Fig 1d). Cell death does not seem to contribute to the production of proapoptotic paracrine signals, since, BAX/BAK DKO are at least as efficient as control cells in inducing them, despite their robust resistance to paclitaxel-induced cell death.

Rebuttal Figure 2: Cell death analysis was performed after a 48h paclitaxel treatment in MDA-MB-468 control or BAX/BAK double KO cells.

Serum deprivation does arrest cells in G1, but does a lot more to cells than simply blocking their proliferation. Hence, it is not entirely clear whether the transcriptional/translational capacity of such cells (following serum starvation) affects the conclusion. Cells should be arrested in other ways too.

We thank the reviewer for this suggestion. We found that cell cycle blockade in G1/S phase by thymidine before paclitaxel treatment also prevents paclitaxel-induced mitotic arrest and paclitaxel-driven proapoptotic paracrine effect (included in the manuscript in new Supplementary Figure 1m). This is consistent with the observation we made using SVF deprivation, and brings further support to the notion that progression through mitosis, and mitotic damage are crucial for paclitaxel to trigger the cell inflammatory response.

The effect of knocking out STING or IFNAR1 is modest, even though the authors describe it as ‘abrogate’ the sensitivity. ‘Abrogate’ is clearly the wrong term here. What is the effect of IFNAR1/TNFR1 double knockout cell lines?

We modified this overstatement in the manuscript in the results section in paragraph “Type I IFN and TNF are major contributors of paclitaxel-induced STING-dependent pro-apoptotic paracrine that induce NOXA in recipient cells”: “Either manipulation alone inhibited paracrine apoptotic priming by paclitaxel (Fig 4f-g and supplementary Fig 4c-d). Their combination did not provide further protection, arguing that IFN and TNF act in concert to promote apoptotic priming (supplementary Fig. 4e)”. In our previous version, we had tested separately the effects of media conditioned by paclitaxel-treated TNF^{-/-} donor cells on control cells or the effects of media conditioned by control paclitaxel-treated donor cells on IFNAR1^{-/-} recipient cells, and showed that apoptotic priming was significantly decreased compared in either setting (shown in Fig 4f-g and Supplementary Fig 4c-d). We have now

added in Supplementary Fig 4f, data using in the same experimental settings both TNF^{-/-} donor cells and IFNAR1^{-/-} recipient cells. Results indicate that blocking both TNF and type I IFN pathways was no more efficient to protect cancer cells from apoptosis than blocking either pathway alone. This argues for a synergistic effect between both cytokines to promote apoptotic priming upon paclitaxel treatment, as underscored by experiments using recombinant cytokines separately or combined (see Fig 4i and Supplementary Fig 4g).

Also, it might be worthwhile to repress NF-κB (on its own and in combination with loss of STING) via expression of the NF-κB super-repressor in 'doner' cells to evaluate contribution of NF-κB versus interferon. It is well known that paclitaxel potently triggers NF-κB activation.

We thank the reviewer for this relevant and fruitful suggestion. As recommended, we generated NF-κB repressor cell lines (using the plasmid IκBαSR (S32A/S36A-mutated IκBα (Addgene #15291) coding for an IκBα mutant resistant to phosphorylation and degradation) and used them as donor cells. The results obtained, shown in the new Supplementary Figure 4e) led us to conclude that NF-κB activity is necessary to obtain the optimal apoptotic effect triggered by paclitaxel-treated CM.

4) How does the combination of INFα and TNF drive NOXA accumulation?

This is a really interesting point that we investigated further in this revised version, using Chromatin-IP and siRNA experiments. As described in the text and shown in the new Supplementary Figure 5i, chromatin-IP revealed that IRF3 was detected on the PMAIP1 promoter under both conditions (paclitaxel and TNFα/IFNα treatments) whereas IRF9, which functions more directly downstream of the interferon receptor could not be detected (data not shown). Consistently, down-regulation of IRF3 by siRNA prevented NOXA accumulation and apoptotic priming in cells receiving CM from paclitaxel-treated donors or TNFα/IFNα treatment (Supplementary Figure 5j-k). This assay also shows that slight recruitment of RelA in TNFα/IFNα-treated cells. These data indicate that NOXA induction by TNFα/IFNα ensues from a complex interaction between NF-κB and type I IFN signalling relying on RelA and IRF3. Relevantly, the latter transcription factor was proposed to be a key regulator of IFNα/β receptor-mediated feedforward regulation and crosstalk with other pathways, including NF-κB as described by Wang et al. (J Immunol. 2016 May 15;196(10):4322-30), a reference that we have now included in the manuscript.

5) Figure 6: I am sure that they authors will be aware of the phase I clinical trial with Paclitaxel/Navitoclax in solid tumors that was conducted and prematurely terminated in 2012. The results were disappointing due to significant hematological and non-hematological toxicity [9]. No other phase I trial has been run with this combination since. The protocol used was similar to Protocol #1 (Fig. 6). Protocol #2 similarly uses BH3 mimetics with a toxicity profile that is likely to be similar to the one of the clinical trial. Given the recent data on Venetoclax, it might be worthwhile to explore this BH3 mimetic in combination with paclitaxel, as this is trialled at present in breast cancer [10, 11].

We thank this reviewer for suggesting to discuss how our results fit in the context of past and current clinical data. This aspect is now included in our Discussion section, when we comment data shown in the new Figures 1f and 7f) in the light of the limiting side effects combinations between anti-mitotic agents and BH3 mimetics might have.

A first study evaluating Navitoclax in combination with docetaxel for the treatment of solid tumors hinted that febrile neutropenia was the most commonly observed dose-limiting toxicity for navitoclax (Puglisi M *et al*, A phase I safety and pharmacokinetic (PK) study of navitoclax (N) in combination with docetaxel (D) in patients (pts) with solid tumors. *J. Clin. Oncol.* **29**, 2518 (2011). A subsequent study assessed the safety and pharmacokinetic interactions between Navitoclax and carboplatin/paclitaxel or paclitaxel alone in solid tumors and reported limiting non hematological and hematological toxicities indeed, the latter including neutropenia but also thrombocytopenia (Vlahovic G *et al*. A phase I safety and pharmacokinetic study of ABT-263 in combination with carboplatin/paclitaxel in the treatment of patients with solid tumors. *Invest New Drugs* **32**, 976-984

(2014). These data indicate that the use of more selective BH3 mimetics may be useful by reducing side effects. Venetoclax, which shows rationalized and promising results in the context of endocrine therapy of breast cancer (Vaillant F *et al.* Targeting BCL-2 with the BH3 mimetic ABT-199 in estrogen receptor-positive breast cancer. *Cancer Cell* **24**, 120-129 (2013) and Lok SW *et al.* A Phase Ib dose-escalation and expansion study of the BCL2 inhibitor venetoclax combined with Tamoxifen in ER and BCL2-positive metastatic breast cancer. *Cancer Discov.* **9**, 354-369 (2019) could be an option.

In this new version of the manuscript, we thoroughly compared the effects of distinct BH3 mimetics on recipient cells exposed to media conditioned by paclitaxel-treated cells (shown in new Figure 1f). These data show that blocking BCL-xL is the only efficient way to increase paclitaxel proapoptotic paracrine. In contrast to WEHI-539, ABT-199 (Venetoclax) or the selective MCL-1 S63845 were inefficient in this setting. In addition, ABT-199 added to organoids obtained from the PDX#306, did not lead to synergistic effect upon to paclitaxel in contrast ABT-737 (these data are included in the revised version of our manuscript in Fig 1j).

These data argue that selective inhibition of BCL-2 may not efficiently exploit paclitaxel-induced proapoptotic paracrine. A contrario, they put forth that a selective BCL-xL inhibitor might be efficient. Thus, they advocate for a combination between anti-mitotic drugs and selective BCL-xL that may have reduced effects on neutropenia (as this side effect was ascribed to BCL-2 inhibition, Levenson JD *et al.* Exploiting selective BCL-2 family inhibitors to dissect cell survival dependencies and define improved strategies for cancer therapy. *Sci Transl Med.* **18**, 279ra40 (2015)) while retaining the effects on thrombocytopenia (which was ascribed to BCL-xL inhibition and accounts for limiting toxicity of navitoclax as a single agent (Gandhi L *et al.* Phase I study of Navitoclax (ABT-263), a novel Bcl-2 family inhibitor, in patients with small-cell lung cancer and other solid tumors. *J Clin Oncol* **29**, 909-916 (2011).

The necessity to target BCL-xL to enhance paclitaxel-induced pro-apoptotic paracrine further justifies our proposal that a better timing in drug administration might improve the efficacy and the safety of the combination. Our data suggest that sequential protocol starting with paclitaxel might allow accumulation of mitotic damaged cancer cells in which activation of cGAS/STING pathway can propagate apoptotic waves in tumor via the active elaboration of a proapoptotic inflammatory secretome, predisposing tumors to the effects of BCL-xL inhibitors administered in a delayed manner. This might open the avenue to decreased BH3 mimetic doses to optimal anticancer effects and possibly to minimal side effects.

It is difficult to conclude that Protocol 1 is worse than protocol 2. Obviously Navitoclax has a particular in vivo half-life. In protocol 2 it seems that the authors administered Navitoclax in a way that maintains a certain level of this drug throughout the treatment, while in protocol one there is a Navitoclax drug holiday. Without a proper characterisation of the kinetics of TNF/IFN induction and NOXA upregulation in vivo, it is difficult to make a strong conclusion.

We thank this reviewer for pointing this out. In the synchronous protocol #1, Navitoclax was administered in D0, D1 and D7, D8 and in the sequential protocol #2, Navitoclax was administered in D3, D5 and D10, D12. There is therefore only one day difference in the Navitoclax drug holiday between protocol #1 (6 days) and protocol #2 (5 days). The fact that a delayed protocol was more efficient to promote cell death in the cancer cell population than a synchronous one was further confirmed when we studied *in vitro* paclitaxel response of organoids obtained from one of our PDX (new Supplementary Fig 7d, note that only one dose of paclitaxel and Navitoclax were used in this setting). Finally, while the *in vivo* protocol #2 did not provide sufficient material for molecular analysis, we could nevertheless extract mRNA from tumors after protocol #1 and investigate the effect of (co)treatments on TNF and NOXA expression. As shown in the Rebuttal Figure 3 below, BH3 mimetic co-treatment prevented both inductions by paclitaxel. This is consistent with the notion that protocol #1 is not fully efficient because co-administration of Navitoclax prevents the mounting of a proapoptotic secretory phenotype by killing too fast secretory cells (which are intrinsically sensitive to Navitoclax as shown in figure 6a).

Rebuttal Figure 3: qPCR analysis of *TNF* and *PMAIP1* in MDA-MB-231 tumors treated in vivo by paclitaxel combined to Navitoclax or not using synchronous protocol # 1

Reviewer #2

STING-dependent paracrine shapes apoptotic priming of breast tumors by anti-mitotic treatment

Here, Lohard et al. investigated the mechanisms of action of the chemotherapeutic drug Paclitaxel (PTX) in breast cancer cells, as well as the potential of combinatorial treatment approaches involving PTX and BH3 mimetics for treatment of breast cancer. Interestingly, they found that STING signaling within the PTX-hit cycling tumor cells aids in apoptotic priming of the neighbouring tumor cells, which are not sensitive to PTX treatment, via the mechanisms regulated by type I IFNs and TNF α produced by PTX-hit tumor cells in a STING-dependent manner. On the other hand, they showed that BH3 mimetics treatment following PTX treatment showed significantly improved anti-tumor effect compared to PTX singular treatment or simultaneous PTX and BH3 mimetics treatment strategies in a STING-dependent manner, emphasizing in vivo relevance of their in vitro data using breast cancer cells from human patients. Therefore, these data demonstrated that such combinatorial therapies might be advantageous compared to singular PTX or BH3 mimetics treatment options only if the treatment protocol is carefully optimized.

We thank this reviewer for his comments.

Major Comments:

1- Here, authors used a combination therapy involving PTX and BH3 mimetics to improve anti-tumor effect of PTX in a murine breast cancer model. Since both PTX and anti-apoptotic therapy could have some serious side effect due to targeting of the cells other than cancer cells in the body, doesn't the authors think that it would be better to combine STING agonists with PTX instead of BH3 mimetics? What do you think are the advantages of PTX + BH3 mimetics combination therapy over PTX + STING agonist combination therapy?

We are grateful to this reviewer for raising this point and for giving us the opportunity to include innovative data using STING agonists. Our mechanistic studies indicate that STING agonists might mimic paclitaxel paracrine effects leading to tumor sensitization to BH3 mimetics. Accordingly, we found that the cyclic nucleotide cGAMP added on organoids obtained from 3 fresh surgical breast cancer specimen recapitulated paclitaxel ability to enhance ABT-737 efficiency leading to a significant synergy between this compound and the BH3 mimetic ABT-737 (data now added in the new Fig 1kj).

For more in depth studies, we turned to the synthetic small molecule diABZI-STING agonist-1, recently published and now commercially available, that is 400-fold more potent than cGAMP to activate STING and to induce secretion of IFN β (Ramanjulu JM *et al*, *Nature* **564**:439-443 (2018)). This compound faithfully recapitulated paclitaxel proapoptotic paracrine effects (induction of a TNF/IFN-dependent

secretome leading to NOXA induction) in recipient breast cancer cells lines by on-target (STING KO sensitive) effects leading to enhanced sensitization to BCL-xL but not BCL2 inhibition using WEHI-539 or ABT-199, respectively. In contrast, diABZI-treated cells were not sensitized to paclitaxel induction of cell death, indicating that direct STING activation might advantageously substitute for paclitaxel in the context of BH3 mimetic treatment, but ruling out that it might substitute for BH3 mimetics (and BCL-xL inhibitors in particular) in the context of anti-mitotic treatments. Importantly, diABZI showed potent antitumor *in vivo* activity in combination with the BH3 mimetic Navitoclax. All these data are now included in the new Fig 7f-k and Supplementary Fig 7e-f.

We also tried to use the STING inhibitor H-151 described in Haag SM *et al* (Targeting STING with covalent small-molecule inhibitors, *Nature* **559**, 269 (2018)) in an attempt to block paracrine effect of paclitaxel. This molecule however appeared to be toxic for breast cancer cells even at the low dose (1 μ M) used in the above mentioned report, in a STING-independent way since even STING KO cells were impacted. We therefore stopped experiments using this compound.

2- In this study, authors are mainly using annexin V staining for measuring apoptotic cell death. However, as other types of cell death, such as necrosis, can also be detected by annexin V staining, it is better to include one more definitive staining, such as propidium iodide staining, which can discriminate between apoptotic and necrotic cells. Regarding this information, do the authors think that PTX induces tumor cell death via apoptosis but not via necrosis or other types of cell death in their experiments?

We have now added in the manuscript experiments of cell death in the presence of the pancaspase inhibitor Q-VD-OPh (Supplementary Fig 1d) showing that Annexin V positive cells completely disappeared in presence of the caspase inhibitor, arguing for an apoptotic process. This corroborates the absence of cell death when using BAX/BAK double KO as shown in the new Figure 6a.

3- In figure 2j and supplementary figure 2f, IFN α production in the untreated STING KO MDA-MB-468 cells is higher compared to WT cells. Is this a significant increase and do the authors have any explanation for this? Is this due to the method used for preparation of these STING KO cells (CRISPR-Cas9 method)? If so, do you think that it would affect the interpretation of the current data?

We noticed this point which seems to be significant. However we do not currently have any explanation for this observation. It is important to note that both control and STING^{-/-} cell lines underwent the same process of lentiviral infection/selection in same time and settings. This point certainly deserves further investigation to better understand the role of STING in cancer cells.

4- Do the authors think that this STING-dependent paracrine mechanisms of action of PTX revealed in breast cancer cells apply for other types of cancer cells, for which PTX therapy is being used in the clinic (e.g. ovarian or non-small cell lung cancer)? In other means, is this STING-dependent paracrine mechanism a general mechanism by which anti-mitotic drugs work in the cancer cells? Have the authors tested other tumor cell types or anti-mitotic drugs for answering this question?

We tested the antimitotic Aurora-B kinase inhibitor AZD1152 and found it to induce cGAS-positive micronuclei and to induce apoptotic priming revealed by enhanced cell death rates upon addition of the BCL-xL inhibitor. This effect was also dependent on STING since it was significantly decreased in STING KO cells (this was now included in the new Supplementary Fig 1o). In contrast we could not detect such effect when cells were treated with the genotoxic agent etoposide (this was now included in supplementary fig 1p).

As suggested, we also used the non-small cell lung cancer cell line A549 and the ovarian cancer cell line SK-OV-3, and provided evidence for the same paclitaxel-induced proapoptotic paracrine effect described in breast cancer cells (this was now included in supplementary Fig 1f-g).

These data indicate that the paracrine proapoptotic effect we report for paclitaxel could be extended to other antimitotic drugs and probably to some other solid tumors.

5- In this study, authors show that PTX acts on tumor cells to activate STING pathway most likely via mitotic arrest-dependent micronuclei formation and subsequent leakage of those micronuclei-bound nucleic acids into the cytosol to activate cGAS/STING pathway. Regarding those nucleic acids and their sensors, do the authors know whether those nucleic acids are composed of DNA-only (or RNA or DNA/RNA hybrids)? Also, is cGAS the only cytosolic nucleic acid sensor responsible for detection of these nucleic acids (what about AIM2 or others) in these experimental settings?

We thank this reviewer for raising this important point. To specify the sensor pathway(s) involved in the proapoptotic paracrine effect of paclitaxel by functional experiments, we produce cell lines KO for the cytosolic DNA sensor cGAS. Cells were then challenged by paclitaxel *in vitro* and *in vivo*. Results clearly indicate that cGAS was a critical sensor upstream of STING to trigger pro-apoptotic paracrine and tumor response. These results are now included in new figures Fig 1b, Supplementary Fig 1b, 1d, 4c and 7c. They argue that cytosolic DNA is the predominant molecular pattern involved in (LMB2 overexpression-sensitive) paclitaxel induced pro-apoptotic paracrine.

Minor Comments:

1- English editing is recommended for this manuscript, especially for discussion section. It is difficult to understand the meaning of the sentences.

We went through extensive editing after reorganizing the outline of the manuscript as suggested by Reviewer 1. We hope that the resulting manuscript is of improved clarity.

Reviewer #3

This manuscript describes the role of paclitaxel-induced extrinsic as well as intrinsic signals in priming breast cancer cells to BH3 mimetic -induced cell death. The study is very well-conducted uses human samples manipulated ex vivo, PDXs and genetically engineered cell line models. Generally, the data presented is very convincing and manuscript proposes a novel and interesting mechanism that impinges on modulating sensitivity to paclitaxel to improve patient outcome. In my opinion, the study does achieve high-level of novelty and conceptual impact one expects

We thank this reviewer for his supportive comments of our results.

Reviewers' comments:

Reviewer #1 (Remarks to the Author):

The revised version of the manuscript by Lohard et al. shows significant improvements and successfully answers several of the key comments. The new organisation of the text and discussion greatly facilitate the accessibility of the findings. We particularly acknowledge the effort undertaken by the authors to improve the ms.

We would like, however, to put forward some additional comments. We think they might further improve some potential weaknesses.

1) Figure 2g and Figure 3 are the weakest point of the manuscript. Figure 2g present fold of increase >2 in pg/mL for IFNa. Later in the paper, the author use 2000UI/uL of IFNa in their experiment. This would correspond to a high ng/mL range depending on the specific activity of the batch. Would IFNb have given better results in term of fold of increase and amount? In Figure 2b, PDX#306 and #248 do not show any interferon signature. PDX#249 shows an interferon type II signature not type I. To close the debate on how much interferon is needed, it would be useful to select and present a panel of ISGs from their DGE-RNAseq to show activation of the IFN pathway. The same thing can be done with a panel of NF-kB responsive genes.

2) Figure 1l left panel is quite illegible and seem to expose an issue with the experimental design. Indeed, the H2B-RFP low quadrants show an abnormally large amount of cell in Annexin V positive cells after WEHI-539, compared to what was available in the Annexin V negative in untreated condition. A corresponding loss can be seen in RFP positive cells. This suggests that RFP-positive cells might be losing RFP while dying and therefore this would greatly bias the conclusions. A reverse experiment by tagging the donor cells might have been useful.

3) Supplementary Figure 1n can only be properly interpreted with controls such as normal cell cycle profile (not adding CM) and positive control of paclitaxel-treated showing the arrest.

4) The result paragraph describing the results shown in Figure 2 over-uses the word "acute" to describe the sensitivity to paclitaxel. This could be misleading. The authors are using 700nM of paclitaxel (Mat&Met Preclinical breast cancer ex-vivo assay). The authors should be aware that the clinical relevant does (the dose that can be achieved in a patient) ranges between 2 to maximal 10 nM. The meaning of the label of "sensitive" already suggest death to some extent. The mention of "acute" should be removed.

5) In a response to one of our comments the authors made use of Etoposide. However, Etoposide does activate STING in certain settings (Dunphy et al, Mol Cell 2018). A western blot confirming that Etoposide does not activate the STING pathway in the cell types used would be important.

Minor comments:

1) The introduction is fairly long and could profit from shortening. More generally, long convoluted sentences should be avoided.

2) Figure 4e, it would be useful to have a validation of STING agonist activity e.g. western blot for pSTAT1 and pRELA.

3) Figure 5f, why does NOXA signal disappear with the single treatments?

4) Supplementary Figure 6b, there is a repetitive glitch on printing showing a background colour demarcation between the first line and the rest

5) The authors have been asked by us and Reviewer #2 to consider the role of programmed necrosis and be aware that Annexin V cannot detect necrotic cells and that the use of PI should be warranted. The fact that Annexin V staining disappears with QVD is only logical and is more of an internal control of the experiment. BAX^{-/-} BAK^{-/-} rescue Annexin V staining, this is again pure logic but does not exclude necrosis. However, the presented data are clearly pointing out to an apoptotic phenomenon and we would understand if the authors were to consider programmed necrosis as being out of the scope of the present manuscript. We would advise for future work to always include PI staining along with Annexin V.

6) The fact that TNF^{-/-} donors and IFNAR^{-/-} recipients still show some death is interesting. Have the authors considered a role for other IFN-inducible death ligands such as TRAIL?

7) Page 4 at the bottom of the page "the canonical STING agonist cGAMP". Canonical cGAMP is 3'3'cGAMP and the non-canonical is 2'3'cGAMP. It is not said in the Mat&Met which one was used but this designation can be confusing for non-experts. "Natural STING ligand" or "natural STING agonist" might be preferred.

Reviewer #2 (Remarks to the Author):

The authors addressed the reviewer's comment properly, by which the current manuscript is much improved.

Rebuttal letter NCOMMS-19-06074B

Reviewer #1

The revised version of the manuscript by Lohard et al. shows significant improvements and successfully answers several of the key comments. The new organisation of the text and discussion greatly facilitate the accessibility of the findings. We particularly acknowledge the effort undertaken by the authors to improve the ms.

We thank this reviewer for his positive comments.

We would like, however, to put forward some additional comments. We think they might further improve some potential weaknesses.

As suggested, we have now completed this re-revised version with additional data or controls reinforcing our work.

1) Figure 2g and Figure 3 are the weakest point of the manuscript. Figure 2g present fold of increase >2 in pg/mL for IFN α . Later in the paper, the author use 2000UI/uL of IFN α in their experiment. This would correspond to a high ng/mL range depending on the specific activity of the batch. Would IFN β have given better results in term of fold of increase and amount?

Figures 2 and 3 describe investigations of human samples that hinted on TNF and IFN-I as plausible actors of the pro-apoptotic paracrine effects of paclitaxel, while further figures establish a causal link between these cytokines inducing PMAIP1/NOXA expression and paracrine. We agree that the concentrations of IFN α we used in these experiments were higher than those detected in tumor supernatants but recombinant cytokines are often less active than natural ones. Even though IFN β may be produced in the same time as IFN α , we only had the opportunity to measure production of the latter by ELISA in tumor supernatants but, due to the scarcity of biological material, not that of the former. Our results therefore do not rule out that IFN α and IFN β synergize upon paclitaxel treatment. Importantly, in this work, our goal was not to decode which IFN-I and which of their gene targets would be specifically induced after paclitaxel treatment (even though these points are biologically relevant). We reasoned instead with global IFN-I effects, therefore using IFNAR1 KO cell lines to block both the effects of IFN α and β . Further investigations specifically focused on IFN-I should be performed to dissect more precisely these points. However, as described below, we deepened our transcriptomic analysis of PDX treated *in vivo* regarding IFN-I signature induction.

In Figure 2b, PDX#306 and #248 do not show any interferon signature. PDX#249 shows an interferon type II signature not type I. To close the debate on how much interferon is needed, it would be useful to select and present a panel of ISGs from their DGE-RNAseq to show activation of the IFN pathway. The same thing can be done with a panel of NF- κ B responsive genes.

In Figure 3b (as the reviewer probably meant), we included only the six first (most significant) signatures proposed by EnrichR analysis of rather parcimonious DGseq data. To address this reviewer's concern, we completed the analysis with new qPCR experiments to measure ISGs and NF- κ B responsive genes. These data, now added in supplementary Figure 3c, indicate that 4 ISG (IFIT1, OASL, MX1 and STING) were significantly upregulated in the paclitaxel-treated PDXs (with the exception of IFIT1 which only tended to be increased in PDX #248). Moreover, expression of both BIRC2 and BIRC3 NF- κ B target genes was also increased in PDX #249 and #306 paclitaxel-treated tumors. This was not detected in PDX #248 despite induction of TNF (Figure 3c) and of PMAIP1/NOXA (Figure 5c) indicating a specification of TNF signaling in this tumor that requires further investigation. All of these new data are now included in supplementary Figure 3c.

Importantly, these transcriptomic exploratory data regarding IFN-I and TNF pathways activation in PDX after *in vivo* paclitaxel treatment were further confirmed in the manuscript by functional experiments in cancer cell lines. We feel that these additional data now correctly answer to the reviewer's request.

2) Figure 1l left panel is quite illegible and seem to expose an issue with the experimental design. Indeed, the H2B-RFP low quadrants show an abnormally large amount of cell in Annexin V positive cells after WEHI-539, compared to what was available in the Annexin V negative in untreated condition. A corresponding loss can be seen in RFP positive cells. This suggests that RFP-positive cells might be losing RFP while dying and therefore this would greatly bias the conclusions. A reverse experiment by tagging the donor cells might have been useful.

The goal of this figure was to investigate whether cells that had not been exposed to paclitaxel (marked with H2B-RFP) would nevertheless undergo cell death when treated with a BCL-xL inhibitor in the presence of cells that had been pre-exposed (unlabeled). Reading of Annexin V positivity in H2B-RFP high quadrants (that is, in cells that patently retained RFP) indicate that this is the case, provided that the unlabeled donor cells express STING. In this experimental design, unlabeled donor cells (H2B-RFP low quadrant, wild type or STING KO) pre-exposed to paclitaxel were *de facto* treated with WEHI-539 (upon their co-culture with H2B-RFP positive cells). The rates of cell death obtained in these cells should therefore correspond to what we described further in the manuscript when we explored the direct, intrinsic BCL-xL priming effects of paclitaxel and showed them to be STING independent (Supplementary Figure 6A). Cell death rates are comparable in the H2B-RFP low quadrants and in Supplementary Figure 6A.

Since we did not generate the STING KO expressing H2B-RFP, we could not perform the reverse experiment using H2B-RFP donor cells as proposed by the reviewer. We nevertheless tested whether the RFP staining (linked to the histone H2B and nuclear) resisted to apoptosis induction by treating H2B-RFP positive wild type cells with paclitaxel and WEHI-539. As shown below, the vast majority of cells retained RFP staining under these conditions, arguing that RFP staining is sufficiently robust to allow cell death analysis in specific cell populations (see Rebuttal Figure 1 below).

We improved the figure typographic legibility in % quadrants.

Rebuttal Figure 1

3) Supplementary Figure 1n can only be properly interpreted with controls such as normal cell cycle profile (not adding CM) and positive control of paclitaxel-treated showing the arrest.

The supplementary figure has now been completed with requested controls.

4) The result paragraph describing the results shown in Figure 2 over-uses the word “acute” to describe the sensitivity to paclitaxel. This could be misleading. The authors are using 700nM of paclitaxel (Mat&Met Preclinical breast cancer ex-vivo assay). The authors should be aware that the clinical relevant does (the dose that can be achieved in a patient) ranges between 2 to

maximal 10 nM. The meaning of the label of “sensitive” already suggest death to some extent. The mention of “acute” should be removed.

To avoid any confusion, the mention of « acute » has been removed. We would also like to point out that, as discussed by Beth A Weaver, paclitaxel accumulates in cancer cells by 50 to more 1000-fold (depending on cell type) compared to plasma leading to intratumoral concentrations from 1 to 9 μM (Weaver BA, Mol Biol Cell. 2014 Sep 15; 25(18): 2677–2681, Zasadil *et al*, Sci Transl Med. 2014 Mar 26; 6(229): 229ra43). The concentration of 700nM thus remains clinically relevant.

5) In a response to one of our comments the authors made use of Etoposide. However, Etoposide does activate STING in certain settings (Dunphy et al, Mol Cell 2018). A western blot confirming that Etoposide does not activate the STING pathway in the cell types used would be important.

We evaluated etoposide effects on IFN-I or TNF pathways using qPCR analysis and evidenced that etoposide treatment did not induce IFNB1 or TNF gene transcription in contrast to paclitaxel in MDA-MB-468 (Rebuttal Figure 2). This coincides with the absence of significant proapoptotic effect of CM from etoposide-pretreated cells (Figure 1p), arguing for specific activity of antimetabolites in this observation.

Rebuttal Figure 2

Minor comments:

1) The introduction is fairly long and could profit from shortening. More generally, long convoluted sentences should be avoided.

We carefully addressed this point and we hope that the resulting manuscript is of improved clarity.

2) Figure 4e, it would be useful to have a validation of STING agonist activity e.g. western blot for pSTAT1 and pRELA.

We validated that the agonist activity of the STING agonist DiABZi, using qPCR analysis that revealed that this compound did activate STING pathway since it induced the transcription of ISGs in MDA-MB-468 cell line (see Rebuttal Figure 3 below).

Rebuttal Figure 3

3) Figure 5f, why does NOXA signal disappear with the single treatments?

While levels of NOXA protein are usually barely detectable in control conditions, NOXA protein expression happens to appear in lysates from untreated cells. This may be due to increased cell density, and it may be less likely to occur when cells are washed and treated with TNF α or IFN α alone that have no effect on NOXA mRNA (Figure 5f). In contrast, NOXA expression was significantly enhanced when cancer cells were treated with both cytokines (IFN α + TNF α).

4) Supplementary Figure 6b, there is a repetitive glitch on printing showing a background colour demarcation between the first line and the rest

The Supplementary Figure 6a (we supposed) has now been improved.

5) The authors have been asked by us and Reviewer #2 to consider the role of programmed necrosis and be aware that Annexin V cannot detect necrotic cells and that the use of PI should be warranted. The fact that Annexin V staining disappears with QVD is only logical and is more of an internal control of the experiment. BAX $^{-/-}$ BAK $^{-/-}$ rescue Annexin V staining, this is again pure logic but does not exclude necrosis. However, the presented data are clearly pointing out to an apoptotic phenomenon and we would understand if the authors were to consider programmed necrosis as being out of the scope of the present manuscript. We would advise for future work to always include PI staining along with Annexin V.

We are confident that cell death we observed was indeed apoptosis based on reviewer's arguments. However we also used PI in addition to AnnexinV in some experiments and we did observe AnnexinV positive but PI negative cells, characterizing an apoptotic process.

Rebuttal Figure 4

6) The fact that TNF $^{-/-}$ donors and IFNAR $^{-/-}$ recipients still show some death is interesting. Have the authors considered a role for other IFN-inducible death ligands such as TRAIL?

TRAIL has not been studied in this work but would be interesting to evaluate in another study. We thank the reviewer for this comment.

7) Page 4 at the bottom of the page "the canonical STING agonist cGAMP". Canonical cGAMP is 3'3'cGAMP and the non-canonical is 2'3'cGAMP. It is not said in the Mat&Met which one was used but this designation can be confusing for non-experts. "Natural STING ligand" or "natural STING agonist" might be preferred.

This point has now been modified in the newly revised manuscript.

Reviewer #2

The authors addressed the reviewer's comment properly, by which the current manuscript is much improved.

We thank this reviewer for his supportive comment.